# What Do Agents Learn from Trajectory-SFT: Semantics or Interfaces?

Weizheng Gu [* 1]  Chengze Li [* 2]  Zhuohao Yu [* 1]  Mengyuan Sun [1]  Zhibang Yang [1]  Wei Wang [3]  Hongrui Jia [1]  Shikun Zhang [1]  Wei Ye [1]

## Abstract

Large language models are increasingly evaluated as interactive agents, yet standard agent benchmarks conflate two qualitatively distinct sources of success: semantic tool-use and interface-specific interaction pattern memorization. Because both mechanisms can yield identical task success on the original interface, benchmark scores alone are not identifiable evidence of environment-invariant capability. We propose **PIPE**, a protocol-level evaluation augmentation for diagnosing interface reliance by minimally rewriting environment interfaces while preserving task semantics and execution behavior. Across 16 environments from AgentBench and Agent-Gym and a range of open-source and API-based agents, PIPE reveals that task-specific trajectory-SFT can amplify reliance on training-time interface forms: in several environments, agents with trajectory-SFT degrade sharply under minimal interface rewrites, whereas other agents are often more stable. We further introduce Interface Reliance (IR), a counterbalanced alias-based metric that quantifies preference for training-time interfaces, and show that interface shortcutting exhibits environment-dependent, non-monotonic training dynamics that remain invisible under standard evaluation. Our code is available at Here.

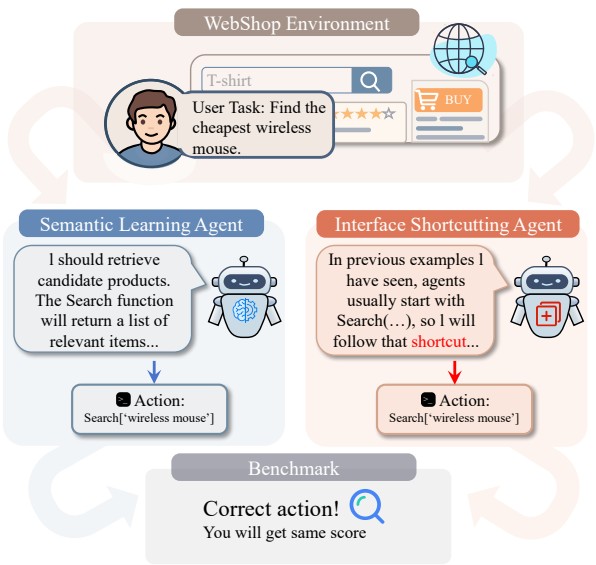

*Figure 1.* Identical scores do not imply identical agent capabilities.

tion answering(Wang et al., 2023; Ho et al., 2020; Yu et al., 2024; Trivedi et al., 2022), agentic tasks require a model to select environment-provided actions under evolving states and to adapt its behavior based on intermediate observations (Wu et al., 2025b; Team et al., 2025b; Mohammadi et al., 2025). Accordingly, agent performance is typically evaluated by task success rate on benchmarks (Liu et al., 2023a; Fang et al., 2025; Li et al., 2025).

A widely adopted approach for improving agent performance is *trajectory-based supervised fine-tuning* (trajectory-SFT) (Zeng et al., 2024; Wu et al., 2025a). In this paradigm, a strong teacher model (e.g., GPT-style models) interacts with the environments to produce action trajectories, which are then used to fine-tune a target model (Chen et al., 2024). Trajectory-SFT has been shown to substantially improve benchmark success rates and has become a standard component in modern agent training pipelines (Zhao et al., 2025).

However, additional training can encourage shortcut solutions, increasing reliance on surface patterns that do not reflect robust generalization (Chatterjee & Mishchenko, 2020; Liu et al., 2023b; Springer et al., 2025). This risk is particularly relevant for trajectory-SFT, which is often evaluated un-

## 1. Introduction

Large language models (LLMs) have recently been extended into *agents* that can interact with external environments, invoke tools, and solve tasks through multi-turn interaction (Xi et al., 2025a; Zhang et al., 2026; 2025; Grattafiori et al., 2024; Jiang et al., 2025). Unlike conventional ques-

---

[*]Equal contribution [1]National Engineering Research Center for Software Engineering, Peking University, Beijing, China [2]Nanjing University, Nanjing, China [3]Tongji University, Shanghai, China. Correspondence to: Wei Ye <wye@pku.edu.cn>.

*Proceedings of the 43rd International Conference on Machine Learning*, Seoul, South Korea. PMLR 306, 2026. Copyright 2026 by the author(s).

der protocols that closely match the training interface (Zeng et al., 2024; Chen et al., 2024; Song et al., 2024; Xi et al., 2025b). Trajectory-SFT can improve benchmark scores via two effects: (i) **semantic learning**, where the agent internalizes tool meaning and effects; and (ii) **interface shortcut**, where it reproduces interaction patterns tied to the interface surface form. As Figure 1 illustrates, both can yield identical scores on the original interface, while semantic learning is more likely to support transfer across interface variants. This distinction matters when benchmark scores are used as evidence of transferable agent capability rather than fixed-interface optimization.

To provide a controlled diagnostic for this ambiguity, we propose **Perturb Interface Protocol for Evaluation** (PIPE), which rewrites only the interface (e.g., action names/formats) while keeping tasks and environment dynamics unchanged. By breaking correspondences between training and evaluation interfaces while aiming to preserve task semantics and intrinsic difficulty, PIPE enables a more direct diagnosis of whether an agent relies on interface shortcuts rather than semantic understanding.

These results suggest that, under standard evaluation, observed trajectory-SFT gains should not be automatically attributed to improved semantic capability alone. Case studies further show that trajectory-SFT agents can over-rely on training-time interfaces: when the interface changes, they may persistently attempt to invoke actions using deprecated names or formats. Such failures motivate the need for a quantitative measure of interface dependence. To this end, we introduce a new metric, *Interface Reliance* (IR). Finally, our analyses indicate that interface shortcutting is not inevitable: with appropriate training design, agents can reduce interface reliance and recover semantic learning on some environments.

In summary, our contributions are threefold:

- We reveal a fundamental ambiguity in the evaluation of trajectory-trained agents: under standard benchmarks, improvements in success rates are not sufficient to distinguish **semantic learning** from **interface shortcut**, rendering benchmark scores alone inadequate for interpreting agent capability.

- We introduce PIPE, a protocol-level diagnostic framework that minimally perturbs environment interfaces while preserving task semantics, and define Interface Reliance (IR), a novel metric that quantifies an agent's dependence on training-time interfaces, thereby enabling the isolation and measurement of interface shortcuts on existing benchmarks.

- Through extensive evaluation with PIPE and Interface Reliance (IR), we show that task-specific trajectory-SFT can amplify reliance on training-time interface

forms, with the extent of this reliance varying substantially across environments. These findings highlight the need for interface-aware diagnostics when interpreting agent benchmark gains.

## 2. Related Work

### 2.1. LLM-based Agents and Trajectory-based Tuning

LLM-based agents solve interactive tasks by repeatedly conditioning on observations and invoking environment-provided actions (Du et al., 2025). Early work largely improves agent behavior via prompting and reasoning–action patterns (e.g., ReAct-style prompting) or by building frameworks around strong API-based models (Yao et al., 2022b; Shinn et al., 2023), and has been instantiated in diverse interactive environments such as web navigation and shopping (Deng et al., 2023; Yao et al., 2022a; Tao et al., 2025a).

A dominant approach for improving open-source agents is trajectory-SFT (Chen et al., 2025; Song et al., 2025; Tao et al., 2025b), which collects expert interaction trajectories and trains agents via behavioral cloning, often yielding substantial gains on benchmark success rates (Zeng et al., 2024; Chen et al., 2024; Xi et al., 2025b; Song et al., 2024). However, trajectory-tuned agents exhibit brittle behavior under small interface perturbations or distribution shifts, suggesting potential overfitting to training-time interfaces (Fu et al., 2025). Different from prior work that primarily uses such brittleness to motivate new training paradigms, we study an evaluation question: whether standard benchmarks provide enough evidence to interpret the source of agent improvements.

### 2.2. Evaluation of Agents

Interactive benchmarks typically evaluate agents by task success (or closely related completion metrics), enabling systematic comparisons across prompting and training methods (Yao et al., 2022a; Deng et al., 2023; Liu et al., 2023a; Chen et al., 2026; Xi et al., 2025b; Mohammadi et al., 2025; Yehudai et al., 2025). While effective for measuring *what* an agent accomplishes, success-rate evaluation provides limited evidence about *why* particular tool are selected.

## 3. Perturb Interface Protocol for Evaluation

### 3.1. Formulation of Agent Evaluation Protocol

We formalize agent evaluation at the granularity of an agent solving a single task $\mathcal{T}$ via multi-turn interaction. We denote the collection of environments as $\mathbb{E}$. For a specific environment $\mathcal{E} \in \mathbb{E}$, each task $\mathcal{T}$ is specified by a task prompt $P_{\mathcal{T}}$. Each environment $\mathcal{E}$ provides: (i) an environment description prompt $P_{\mathcal{E}}$, (ii) a set of available actions $\mathcal{A}^{\mathcal{E}} = \{A_i\}$ (tools/functions the agent may invoke), and (iii) an *inter-*

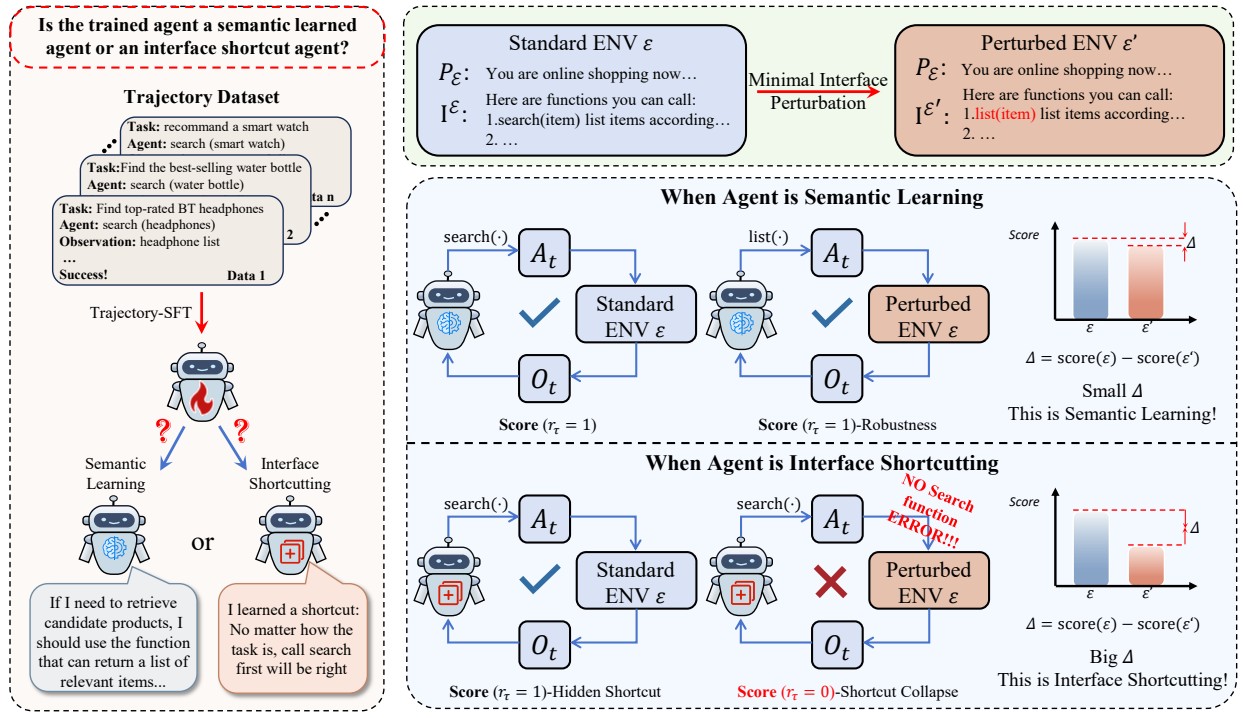

*Figure 2.* How does PIPE distinguish semantic learning and interface shortcutting.

*face specification* $\mathcal{I}^{\mathcal{E}} = \{D_i\}$ for these actions. Each $D_i$ contains the action name, a functional description, and an invocation format (i.e., how the action should be called in text).

Given $(P_{\mathcal{E}}, P_{\mathcal{T}}, \mathcal{A}^{\mathcal{E}}, \mathcal{I}^{\mathcal{E}})$, the agent interacts with the environment for at most $H$ rounds. At round $t$, the agent observes an observation $O_t$ (with $O_0$ possibly empty) and produces an action call $\hat{A}_t$ in text form:

$$\hat{A}_t \sim \pi_\theta\big(P_{\mathcal{E}}, P_{\mathcal{T}}, \mathcal{I}^{\mathcal{E}}, O_{\leq t}\big),$$

where $\pi_\theta$ is the agent policy parameterized by $\theta$. The environment parses $\hat{A}_t$ and executes the corresponding action in $\mathcal{A}^{\mathcal{E}}$ if the call is valid, then returns the next observation $O_{t+1}$. The interaction terminates when the task is solved or the step limit $H$ is reached.

We denote the resulting interaction trajectory as

$$\tau = (O_0, \hat{A}_0, O_1, \hat{A}_1, \ldots, O_L), \quad L \leq H.$$

After termination, the environment assigns an outcome (reward) $r(\tau) \in \{0, 1\}$ indicating task success.[1]

A benchmark contains multiple environments, each with a set of tasks. For an environment $\mathcal{E}$ with task set $\mathcal{T}^{\mathcal{E}}$, the score is the average outcome $r(\tau)$ over tasks in $\mathcal{T}^{\mathcal{E}}$.

Trajectory-SFT datasets are constructed through the same interaction protocol. A strong teacher policy $\pi^\star$ interacts

---

[1]In some environments $r(\tau) \in \mathbb{R}$.

with tasks to generate trajectories $\tau$, which may be further filtered to form a training set $\mathcal{D}$. Benchmark evaluation reuses the protocol but aggregates outcomes $r(\tau)$ as success rates. Therefore, when trajectory collection and evaluation are conducted in the same environment $\mathcal{E}$ with the same interface specification $\mathcal{I}^{\mathcal{E}}$, the fine-tuned agent has been exposed during training to $P_{\mathcal{E}}$, $\mathcal{A}^{\mathcal{E}}$, and crucially $\mathcal{I}^{\mathcal{E}}$. This similarity creates room for trajectory-SFT to improve scores by exploiting interface-level regularities.

### 3.2. Minimal Interface Perturbation

Figure 2 illustrates the evaluation-time augmentation introduced by PIPE. Given an environment $\mathcal{E}$ with interface specification $\mathcal{I}^{\mathcal{E}}$, PIPEconstructs a perturbed counterpart $\mathcal{E}'$ by rewriting *only* $\mathcal{I}^{\mathcal{E}}$ into $\mathcal{I}^{\mathcal{E}'}$. The rewrite changes action names in $\{D_i\}$, while keeping the executable action behaviors, tasks, and observations unchanged.

We consider two perturbation families that differ in whether lexical meaning is preserved:

- **Synonym-based perturbation.** Action names are replaced with semantically related alternatives that retain natural-language meaning.

- **Symbol-based perturbation.** Action names are replaced with semantically meaningless tokens, while keeping detailed functional descriptions in $\mathcal{I}$.

Note that *the agent need not infer tool functionality solely from action names*; each function is accompanied by a detailed description, which is preserved under perturbations. By restricting changes to $\mathcal{I}$, the protocol aims to preserve intrinsic task difficulty and isolate the agent's dependence on interface surface forms.

### 3.3. Distinguish Learning Effects

The similarity between trajectory collection and benchmark evaluation (subsection 3.1) implies that trajectory-SFT can improve benchmark scores through two qualitatively different effects: (i) **Semantic Learning**, where the agent selects and invokes tools based on their functional semantics described in $\mathcal{I}$ and validated by observations; and (ii) **Interface Shortcut**, where the agent exploits surface regularities tied to particular action names or invocation patterns that appeared frequently in training trajectories. Both effects can raise success rates on the original benchmark interface. However, when benchmark scores are used to assess transferable agent capability, interface-specific shortcuts should be distinguished from semantic tool-use.

Let $\mathcal{E}'$ be the perturbed counterpart of $\mathcal{E}$ produced by PIPE, where only the interface specification is rewritten. We evaluate an agent on both $\mathcal{E}$ and $\mathcal{E}'$ and consider the performance gap

$$\Delta(\pi_\theta; \mathcal{E}) = \text{Score}(\pi_\theta; \mathcal{E}) - \text{Score}(\pi_\theta; \mathcal{E}'),$$

where $\text{Score}(\cdot)$ denotes task success rate as defined in subsection 3.1. As shown in Figure 2, a large $\Delta$ indicates stronger dependence on the original interface surface form, consistent with a stronger ISS component. A small $\Delta$ suggests that the agent can better recover tool use from the rewritten interface, which is consistent with weaker dependence on the original surface form.

## 4. Re-Evaluation on Agents

We conduct a re-evaluation of representative trajectory-SFT agents under PIPE. Our goal is to answer two questions: (i) **Q1. Whether the perturbations introduced by PIPE substantially increase the intrinsic difficulty of the tasks**; (ii) **Q2. Whether trajectory-SFT amplifies dependence on training-seen interface realizations, as reflected by larger performance gaps under perturbation**.

### 4.1. Details of Re-Evaluation

We apply PIPE to two widely used agent benchmarks, AgentBench (Liu et al., 2023a) and AgentGym (Xi et al., 2025b), covering a total of 16 environments. Most environments use task success rate as the primary evaluation metric, while a small subset (e.g., Mind2Web in Agent-Bench) adopts benchmark-specific reward signals; in all cases, higher values indicate better performance. Implementation details of the interface perturbations, along with per-environment descriptions and evaluation metrics, are provided in Appendix C. For clarity, we refer to the perturbed benchmarks as **AgentBenchPlus** and **AgentGym-Plus**.

All agents are evaluated under three interface conditions: (i) the original interface, (ii) synonym-based perturbations, and (iii) symbol-based perturbations. For the perturbations, we carefully design action-name aliases and accompanying functional descriptions to ensure that they do not increase task difficulty; details are provided in Appendix I. We report the original score $s_0$, the perturbed scores $s_1$ and $s_2$, and define $\Delta = (s_1 + s_2)/2 - s_0$ as the average performance change induced by interface perturbations.

We first re-evaluate agents trained on trajectories collected from AgentBench (Liu et al., 2023a). AgentInstruct (Zeng et al., 2024) collects interaction trajectories from AgentBench environments and fine-tunes the LLaMA2 (Touvron et al., 2023) family, releasing the AgentLM-7B/14B/70B models. Similarly, Agent-FLAN (Chen et al., 2024) constructs its tuning data from AgentBench interactions and reports substantial performance gains on the original benchmark.

For AgentGym (Xi et al., 2025b), we did not identify prior work that directly fine-tunes agents using its released trajectories. We therefore fine-tune Qwen3-8B/14B (Yang et al., 2025) and Gemma3-4B/12B (Team et al., 2025a) on the AgentTraj dataset provided by AgentGym, and compare performance *before* and *after* training.

Training hyperparameters and additional implementation details are deferred to Appendix K.

Finally, we evaluate API-based models (GPT-3.5(OpenAI, 2023a), GPT-4o-mini(OpenAI, 2023b), and GPT-5-mini(OpenAI, 2025)) to examine whether strong proprietary agents also exhibit sensitivity to the interface perturbations introduced by PIPE .

To avoid ambiguity, we use *non-TrajSFT agents* to denote agents that have not undergone task-specific trajectory-SFT on the evaluated benchmark environments, although they may have undergone pretraining, instruction tuning, or other forms of general post-training. We use *TrajSFT agents* to denote agents that are further fine-tuned on task-specific interaction trajectories from the corresponding benchmark. This terminology distinguishes task-specific trajectory supervision from general instruction tuning or broad post-training, and is used throughout the following results and analyses.

*Table 1.* Performance comparison of non-TrajSFT agents and agents trained with AgentInstruct on AgentBenchPlus across different environments.

| Environment | | API-based Agents | | | Agents after Trajectory-SFT | | | |
| --- | --- | --- | --- | --- | --- | --- | --- | --- |
| | | GPT3.5 | GPT4o-mini | GPT5-mini | AgentLM-7b | AgentLM-14b | AgentLM-70b | AgentFLAN |
| ALFWORLD | Origin | 45 | 45 | 60 | 5 | 40 | 50 | 0 |
| | Perturb 1 | 55 | 25 | 50 | 0 | 30 | 45 | 5 |
| | Perturb 2 | 40 | 40 | 45 | 0 | 0 | 35 | 0 |
| | Δ | 2.5 | -12.5 | -12.5 | -5 | -25 | -10 | 2.5 |
| OPERATINGSYSTEM | Origin | 34 | 21.5 | 29.9 | 9.03 | 14.6 | 22.2 | 6.25 |
| | Perturb 1 | 34.7 | 22.2 | 29.2 | 6.25 | 6.99 | 13.9 | 9.72 |
| | Perturb 2 | 43.8 | 6.94 | 28.5 | 4.17 | 6.94 | 21.5 | 6.94 |
| | Δ | 5.25 | -6.93 | -1.05 | -3.82 | -7.635 | -4.5 | 2.08 |
| DATABASE | Origin | 57.3 | 49.3 | 59 | 24 | 38 | 42 | 28.3 |
| | Perturb 1 | 53.3 | 47.7 | 58 | 31.7 | 38.3 | 2 | 29.3 |
| | Perturb 2 | 55.3 | 49.3 | 58.7 | 27 | 38.3 | 13.3 | 26.3 |
| | Δ | -3 | -0.8 | -0.65 | 5.35 | 0.3 | -34.35 | -0.5 |
| MIND2WEB | Origin | 2.84 | 0 | 0 | 7.95 | 10.2 | 14.8 | 8.52 |
| | Perturb 1 | 9.66 | 16.5 | 0 | 8.52 | 9.09 | 13.1 | 10.8 |
| | Perturb 2 | 7.95 | 2.27 | 0 | 7.39 | 9.09 | 7.39 | 10.2 |
| | Δ | 5.965 | 9.385 | 0 | 0.005 | -1.11 | -4.555 | 1.98 |
| WEBSHOP | Origin | 46 | 48.3 | 36.2 | 55.2 | 61.2 | 50.2 | 46.1 |
| | Perturb 1 | 56.4 | 44.3 | 34.2 | 1.25 | 4.44 | 2.33 | 13.6 |
| | Perturb 2 | 50.5 | 38.3 | 1.25 | 2.4 | 3.08 | 0 | 0 |
| | Δ | 7.45 | -7 | -18.475 | -53.375 | -57.44 | -49.035 | -39.3 |
| KNOWLEDGEGRAPH | Origin | 35.3 | 28.9 | 51.9 | 22.3 | 28 | 35.4 | 2.5 |
| | Perturb 1 | 46.2 | 28.3 | 58.9 | 14.5 | 25 | 25 | 0 |
| | Perturb 2 | 46.9 | 26.3 | 53.2 | 16.8 | 21.4 | 28.3 | 2.5 |
| | Δ | 11.25 | -1.6 | 4.15 | -6.65 | -4.8 | -8.75 | -1.25 |

## 4.2. Difficulty-Control Validation

To directly answer Q1, we conduct two LLM-as-judge difficulty-control experiments to examine whether the interface perturbations introduced by PIPE substantially increase task difficulty. In both experiments, we concatenate the environment description and task description into a full task specification, and ask the evaluator to judge task difficulty under the assumption that the evaluated agent is a competent agent that can read action descriptions. We use `gpt-5.1` as the evaluator. The full prompts and per-environment results are provided in Appendix L.

First, we perform pairwise relative comparisons between the original interface and each perturbed interface. For each pair, the evaluator is asked whether, assuming unchanged backend dynamics, task goals, and action semantics, the interface rewrite makes the task *Easier*, *Same*, or *Harder*. Across all environments in AgentBenchPlus and AgentGymPlus, the evaluator returns *Same* for almost all comparisons. The only minor deviation appears in MOVIE from AgentGymPlus, where the agreement with *Same* remains 95%. This result suggests that the rewritten interfaces do not materially change task difficulty from the perspective of a strong evaluator.

*Table 2.* Average overall difficulty scores from the LLM-as-judge absolute difficulty evaluation. Higher scores indicate greater difficulty. Full per-environment results are provided in Appendix L.

| Benchmark | Origin | Perturb 1 | Perturb 2 |
| --- | --- | --- | --- |
| AgentBenchPlus | 2.44 | 2.43 | 2.54 |
| AgentGymPlus | 3.64 | 3.66 | 3.70 |

Second, we perform absolute difficulty scoring for the original, synonym-based, and symbol-based perturbation separately. The evaluator rates each condition on a 1–5 scale along dimensions including interface clarity, interaction complexity, and overall difficulty, where higher scores indicate greater difficulty.

As summarized in Table 2, the average overall difficulty scores remain close across the three interface conditions. Neither AgentBenchPlus nor AgentGymPlus shows a systematic shift toward higher difficulty under perturbation. Together, these two difficulty-control experiments provide direct evidence for Q1: the interface perturbations introduced by PIPE do not show a systematic tendency to make tasks intrinsically harder.

*Table 3.* Performance comparison of non-TrajSFT agents and agents trained with AgentTraj-L on AgentGymPlus across different environments.

| Environment | | API-based Agents GPT | | | Agents without Trajectory-SFT Gemma3 | | Qwen3 | | Agents after Trajectory-SFT Gemma3 | | Qwen3 | |
| --- | --- | --- | --- | --- | --- | --- | --- | --- | --- | --- | --- | --- |
| | | 3.5-turbo | 4o-mini | 5-mini | 4b | 12b | 8b | 14b | 4b | 12b | 8b | 14b |
| ALFWORLD | Origin | 31 | 27 | 45.5 | 7.5 | 22.5 | 46 | 50.5 | 30.5 | 46 | 51 | 55 |
| | Perturb 1 | 43.5 | 25.5 | 45 | 7 | 25 | 47 | 49 | 10.5 | 31.5 | 36.5 | 44 |
| | Perturb 2 | 47.5 | 21 | 48.5 | 5 | 27 | 52 | 45 | 4.5 | 13 | 37.5 | 36 |
| | Δ | 14.5 | -3.75 | 1.25 | -1.5 | 3.5 | 3.5 | -3.5 | -23 | -23.75 | -14 | -15 |
| BABYAI | Origin | 49.76 | 77.3 | 89.64 | 77.76 | 77.7 | 46.84 | 84.66 | 86.05 | 82.27 | 79.88 | 83.28 |
| | Perturb 1 | 52.69 | 75.32 | 88.39 | 74.45 | 85.89 | 58.17 | 85.46 | 74.3 | 79.42 | 79.57 | 81.29 |
| | Perturb 2 | 65.16 | 79.26 | 82.57 | 75.12 | 84.93 | 71.09 | 77.6 | 78.72 | 76.88 | 84.15 | 75.5 |
| | Δ | 9.17 | -0.01 | -4.16 | -2.97 | 7.71 | 17.79 | -3.13 | -9.54 | -4.12 | 1.98 | -4.89 |
| MAZE | Origin | 72 | 36 | 84 | 8 | 16 | 80 | 100 | 4 | 8 | 16 | 32 |
| | Perturb 1 | 72 | 40 | 80 | 8 | 16 | 64 | 100 | 8 | 16 | 28 | 16 |
| | Perturb 2 | 72 | 40 | 80 | 8 | 8 | 72 | 100 | 4 | 8 | 24 | 28 |
| | Δ | 0 | 4 | -4 | 0 | -4 | -12 | 0 | 2 | 4 | 10 | -10 |
| MOVIE | Origin | 100 | 85 | 95 | 40 | 85 | 95 | 90 | 60 | 15 | 80 | 95 |
| | Perturb 1 | 95 | 90 | 95 | 55 | 85 | 85 | 90 | 45 | 20 | 85 | 90 |
| | Perturb 2 | 100 | 90 | 95 | 75 | 90 | 85 | 85 | 25 | 5 | 85 | 80 |
| | Δ | -2.5 | 5 | 0 | 25 | 2.5 | -10 | -2.5 | -25 | -2.5 | 5 | -10 |
| SCIWORLD | Origin | 66.43 | 49.21 | 62.4 | 48.33 | 38.65 | 60.08 | 54.55 | 82.23 | 84.65 | 86.16 | 81.15 |
| | Perturb 1 | 62.63 | 43.4 | 75.31 | 41.85 | 35.77 | 55.28 | 51.22 | 52.99 | 57.12 | 67.11 | 65.48 |
| | Perturb 2 | 66 | 46.36 | 67.96 | 41.71 | 29.71 | 50.8 | 50.28 | 51.17 | 50.55 | 59.5 | 54.5 |
| | Δ | -2.11 | -4.33 | 9.24 | -6.55 | -5.91 | -7.04 | -3.81 | -30.15 | -30.82 | -22.86 | -21.16 |
| SQLGYM | Origin | 13.5 | 14 | 13.5 | 9 | 13.5 | 14.5 | 8.5 | 5 | 11.5 | 12 | 16 |
| | Perturb 1 | 12.5 | 12.5 | 10 | 1 | 7 | 11.5 | 10.5 | 7 | 9.5 | 12.5 | 17 |
| | Perturb 2 | 5.5 | 13.5 | 14.5 | 0.5 | 11.5 | 2.5 | 10 | 6.5 | 12.5 | 8.5 | 16 |
| | Δ | -4.5 | -1 | -1.25 | -8.25 | -4.25 | -7.5 | 1.75 | 1.75 | -0.5 | -1.5 | 0.5 |
| TEXTCRAFT | Origin | 90 | 65 | 95 | 42 | 63 | 70 | 75 | 54 | 71 | 72 | 73 |
| | Perturb 1 | 87 | 66 | 91 | 43 | 64 | 60 | 68 | 51 | 67 | 61 | 61 |
| | Perturb 2 | 90 | 59 | 94 | 24 | 57 | 63 | 81 | 40 | 63 | 74 | 77 |
| | Δ | -1.5 | -2.5 | -2.5 | -8.5 | -2.5 | -8.5 | -0.5 | -8.5 | -6 | -4.5 | -4 |
| WEATHER | Origin | 70 | 55 | 70 | 10 | 50 | 65 | 75 | 20 | 20 | 40 | 60 |
| | Perturb 1 | 70 | 55 | 70 | 15 | 45 | 65 | 60 | 25 | 10 | 45 | 55 |
| | Perturb 2 | 60 | 55 | 65 | 10 | 65 | 60 | 60 | 5 | 10 | 40 | 45 |
| | Δ | -5 | 0 | -2.5 | 2.5 | 5 | -2.5 | -15 | -5 | -10 | 2.5 | -10 |
| WEBSHOP | Origin | 15.1 | 2.21 | 21.9 | 28.34 | 16.03 | 10.76 | 9.61 | 85.76 | 86.52 | 86.41 | 88.88 |
| | Perturb 1 | 19.75 | 3.17 | 6.68 | 11.57 | 15.58 | 13.29 | 9.39 | 50.57 | 77.97 | 80.43 | 83.97 |
| | Perturb 2 | 13.01 | 3.59 | 7.29 | 16.05 | 10.85 | 14.57 | 9.73 | 74.95 | 76.77 | 84.71 | 81.82 |
| | Δ | 1.28 | 1.18 | -14.92 | -14.53 | -2.82 | 3.17 | -0.05 | -23 | -9.15 | -3.84 | -5.98 |
| WORDLE | Origin | 88 | 16 | 100 | 4 | 44 | 88 | 100 | 0 | 8 | 20 | 32 |
| | Perturb 1 | 100 | 20 | 100 | 0 | 40 | 88 | 96 | 12 | 8 | 12 | 32 |
| | Perturb 2 | 100 | 12 | 100 | 0 | 32 | 96 | 100 | 4 | 4 | 4 | 20 |
| | Δ | 12 | 0 | 0 | -4 | -8 | 4 | -2 | 8 | -2 | -12 | -6 |

### 4.3. Results

**Non-TrajSFT agents provide a behavioral sanity check.**
Beyond the LLM-as-judge difficulty-control evaluation in subsection 4.2, we further examine whether interface perturbations cause uniform degradation for agents that have not been trained on the original interfaces. On AgentBench-Plus (Table 1), non-TrajSFT agents typically exhibit small or mixed Δ across environments, and sometimes even im-prove (e.g., KNOWLEDGEGRAPH and MIND2WEB). Similarly, on AgentGymPlus (Table 3), non-TrajSFT agents have Δ values close to zero on average, with gains and losses depending on the environment; only a few environments show substantial degradation for both TrajSFT agent and non-TrajSFT agents (e.g., TEXTCRAFT). This behavioral pattern is consistent with the difficulty-control results: in most environments, the perturbations do not introduce a

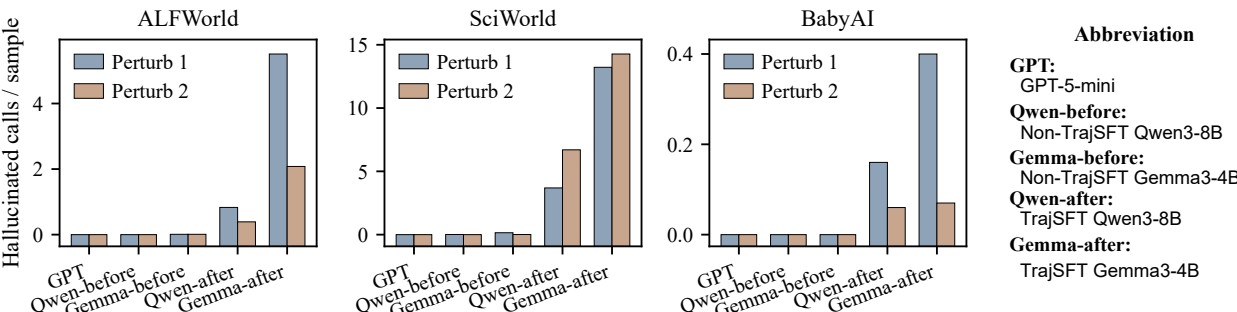

*Figure 3.* Average number of "legacy" action calls per task under interface perturbations on three representative environments.

uniform increase in intrinsic difficulty, and agents can generally re-interpret tool descriptions and adapt to the rewritten interface.

Interestingly, in some cases, non-TrajSFT agents perform better under the symbol-based perturbation than under the original or synonym-based interfaces. One plausible explanation is that, for models without task-specific trajectory training, original action names or their synonyms may activate lexical priors from pretraining that are not always helpful for the current environment. By removing these lexical cues while preserving detailed functional descriptions, the symbol-based perturbation can sometimes encourage the model to rely more on the provided action descriptions rather than superficial word associations. We therefore interpret such improvements not as evidence that symbol-based interfaces are inherently easier, but as further evidence that PIPE acts as a controlled probe of different forms of interface dependence.

**Trajectory-SFT agents are disproportionately brittle under interface perturbations.** In contrast, trajectory-SFT agents exhibit substantially more negative $\Delta$, indicating pronounced sensitivity to minimal interface changes. On AgentBenchPlus (Table 1), `AgentLM` models suffer large degradations in several tool-intensive environments, most notably WEBSHOP (e.g., `AgentLM-14b` drops from 61.2 to 4.44/3.08) and DATABASE (e.g., `AgentLM-70b` drops from 42 to 2/13.3). Notably, `AgentFLAN` is often less affected (e.g., positive $\Delta$ on MIND2WEB and OPERATINGSYSTEM), which is aligned with Agent-FLAN's stated goal of disentangling format following from general reasoning and avoiding overfitting to specific interface (Chen et al., 2024). We emphasize that this is a correlational observation rather than a causal attribution, as we do not perform controlled ablations of Agent-FLAN's training design.

A similar trend holds on AgentGymPlus (Table 3). After fine-tuning on AgentGym trajectories, agents typically achieve clear improvements on the original interfaces (e.g., large gains on WEBSHOP, where TrajSFT agents reach 80–

90+ scores). However, these gains *only partially transfer to* the perturbed interfaces: TrajSFT agents agents frequently incur notable drops on perturbed environments such as ALFWORLD, SCIWORLD, and WEBSHOP, while still often remaining above their non-Trajsft counterparts. This "partial transfer" pattern supports our central claim: trajectory-SFT improves semantic understanding competence, but also induces interface shortcut; PIPE attenuates the latter, revealing that a non-trivial fraction of the original benchmark gains relies on shortcutting.

In a subset of environments, trajectory-SFT agents remain robust to interface perturbations. We provide a detailed environment-factor analysis in Appendix D, which provides additional insights: *To bias trajectory-SFT toward semantic learning (rather than interface shortcuts), our results suggest increasing per-tool supervision: for example, by decomposing complex multi-tool trajectories into simpler, tool-focused trajectories.*

## 5. Failure Analysis under Interface Perturbations

**Q3. Why do agents after trajectory-SFT suffer pronounced performance drops under interface perturbations?** In this section, we answer **Q3** via bad-case studies that highlight failure modes under interface perturbations.

Appendix A presents two representative examples from AgentGym. In WEBSHOP, the training-time action `click` is perturbed to `press`, yet the agent repeatedly outputs `click[...]` despite the environment explicitly restricting valid actions to `find[...]` and `press[...]`. Similarly, in ALFWORLD, the navigation action is changed from `go to <object>` to `navigate to <object>`, but the agent continues to issue `go to <object>` even after receiving repeated invalid-action feedback. Due to space limitation, we provide another bad case and analysis in Appendix B. This behavior suggests that, in these cases, trajectory-SFT agents do not fully rely on the updated tool descriptions when generating actions; instead, they retain

a strong bias toward training-time interface forms. Consequently, even with explicit invalid-action feedback, they often fail to adapt and keep invoking deprecated, training-time action names. To show that this reliance is systematic rather than anecdotal, we quantify such "legacy" action invocations on AgentGymPlus.

Figure 3 highlights the three environments with the most severe interface reliance. TrajSFT agents show a clear tendency to invoke deprecated, training-time action names after perturbation, whereas non-TrajSFT agents virtually never produce such invalid calls. We report the full set of "legacy" action invocation statistics in Table 10.

# 6. Quantifying Interface Reliance

The qualitative analysis in section 5 suggests that agents may over-rely on interfaces encountered during training, leading to brittle behavior when the interface changes. In this section, we try to address **Q4. Is there a principled way to quantify how strongly an agent relies on training-time interfaces?**

We construct modified environments where each underlying action is exposed to the agent through two functionally identical aliases: an *original* interface whose action name matches the training-time interface, and a *synonym* interface whose action name is replaced by a semantically related alternative. Both aliases execute the same underlying operation and therefore differ only in surface form. For implementation details, please see Appendix F.

We quantify an agent's preference for the training-time interface via *Interface Reliance* (IR). Let an environment $\mathcal{E}$ contain tasks $\{\mathcal{T}_k\}_{k=1}^{K}$. For each task $\mathcal{T}_k$, let $n_{\text{ori}}^{(k)}$ and $n_{\text{syn}}^{(k)}$ be the number of times the agent invokes the original and synonym aliases, respectively. We define

$$\text{IR}(\mathcal{E}) = \exp\left(\frac{1}{K}\sum_{k=1}^{K}\log\frac{n_{\text{ori}}^{(k)} + \alpha}{n_{\text{syn}}^{(k)} + \alpha}\right), \qquad (1)$$

where $\alpha > 0$ is a pseudo-count smoothing constant. This is the geometric mean of per-task preference ratios.

To control for positional bias in the ordering of interface descriptions, we use a counterbalanced protocol: we run the full evaluation suite twice (original-first vs. synonym-first), average the per-task $n_{\text{ori}}^{(k)}$ and $n_{\text{syn}}^{(k)}$ across the two runs, and then compute IR from the averaged statistics.

We do not directly use $n_{\text{ori}}^{(k)}/n_{\text{syn}}^{(k)}$ because this ratio can be unstable and overly sensitive to edge cases; a detailed analysis is provided in Appendix G. By computing preference ratios at the task level and aggregating them via geometric averaging, IR mitigates these effects and yields a more faithful measure of interface reliance.

*Table 4.* Interface Reliance (IR) under the counterbalanced dual-interface setting on five AgentBenchPlus environments. *Abbreviations: AW=ALFWorld, DB=DataBase, KG=KnowledgeGraph, OS=OperatingSystem, WS=WebShop.*

| Agent | AW | DB | KG | OS | WS |
|---|---|---|---|---|---|
| *non-TrajSFT Agents* | | | | | |
| gpt-3.5-turbo | 0.96 | 0.41 | 0.87 | 2.72 | 0.99 |
| Qwen3-14b | 1.23 | 0.64 | 0.94 | 0.95 | 0.45 |
| *TrajSFT Agents* | | | | | |
| AgentLM13b | **15.50** | 1.23 | 1.01 | 1.38 | **1.00** |
| AgentLM70b | 9.91 | **2.20** | **1.03** | **3.90** | 0.93 |

*Table 5.* Interface Reliance (IR) under the counterbalanced dual-interface setting on AgentGymPlus environments. *Abbreviations: AW=ALFWorld, BA=BabyAI, MZ=MAZE, SW=SciWorld, WT=Weather, WD=Wordle.*

| Agent | AW | BA | MZ | SW | WT | WD |
|---|---|---|---|---|---|---|
| *non-TrajSFT Agents* | | | | | | |
| GPT-3.5-turbo | 1.03 | 1.32 | 0.92 | 1.08 | 0.97 | 0.96 |
| Gemma3-12b | 0.99 | 1.08 | 0.96 | 0.22 | 0.84 | 0.95 |
| Gemma3-4b | 0.99 | 1.05 | 0.96 | 0.88 | 0.96 | **1.03** |
| Qwen3-14b | 1.02 | 0.98 | 0.98 | 0.79 | 0.85 | 0.94 |
| Qwen3-8b | 1.00 | 0.94 | 1.02 | 0.53 | 1.05 | 0.94 |
| *TrajSFT Agents* | | | | | | |
| Gemma3-12b | 1.75 | **1.80** | 1.15 | 4.20 | **1.69** | 0.97 |
| Gemma3-4b | **4.37** | 1.55 | **1.22** | **7.69** | 0.91 | 0.99 |
| Qwen3-14b | 1.13 | 1.11 | 0.99 | 3.94 | 0.93 | 0.97 |
| Qwen3-8b | 1.42 | 1.22 | 1.01 | 2.65 | 0.95 | 0.97 |

## 6.1. Results on IR.

Table 4 and Table 5 report Interface Reliance (IR) on environments from AgentBenchPlus and AgentGymPlus, respectively. We use $\alpha = 1$ in Equation 1; results with $\alpha \in \{0.5, 2\}$ are deferred to Appendix H and remain highly consistent.

Across both benchmarks, IR exhibits a *split* pattern across environments. In a subset of environments, trajectory-SFT agents show markedly higher IR than non-TrajSFT agents, indicating a strong preference for training-time action names (e.g., DATABASE, OPERATINGSYSTEM, and ALFWORLD in AgentBenchPlus[2]; SCIWORLD and ALFWORLD in AgentGymPlus[3]). In particular, ALFWORLD in AgentBenchPlus demonstrates near-exclusive usage of original action names after training, suggesting extreme interface reliance. In contrast, in another subset of environments IR values cluster around 1 for most agents (e.g., KNOWL-

---

[2]We exclude MIND2WEB because it lacks an explicit action list, making a dual-alias construction non-trivial (see Appendix C).

[3]For AgentGymPlus, we select these environments because they exhibit relatively frequent hallucinated action invocations in the case study presented in section 5, making them particularly informative for analyzing interface reliance.

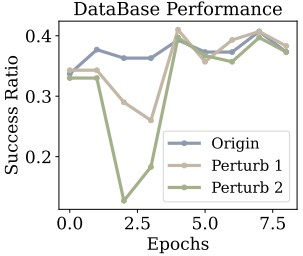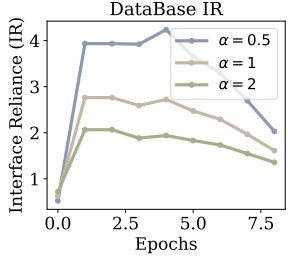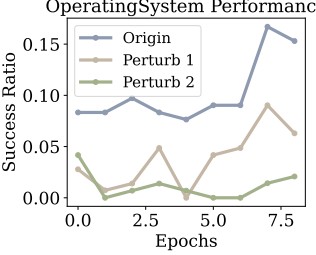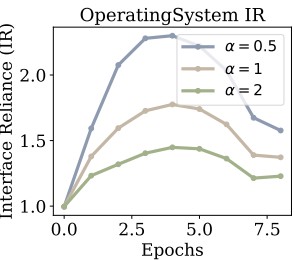

*Figure 4.* Qwen3-8b's performance and IR on DataBase and OperatingSystem across training epochs.

EDGEGRAPH and WEBSHOP in AgentBenchPlus, as well as WORDLE in AgentGymPlus), implying relatively weak preference between original and synonym aliases. These differences suggest that the degree of interface reliance induced by trajectory-SFT is highly environment-dependent: some environments admit strong shortcutting toward surface-form conventions, while others remain largely governed by semantic cues.

## 7. Case Study: Using PIPE and IR as Training-Time Diagnostics

Standard agent evaluation typically monitors only the original-environment success rate during training. In this case study, we show that such a signal can be misleading under trajectory-SFT. By contrast, PIPE and Interface Reliance (IR) provide complementary diagnostics that expose these dynamics and enable more reliable model selection.

We fine-tune `Qwen3-8B` on AgentInstruct for 8 epochs and evaluate each checkpoint on two AgentBenchPlus environments, DATABASE and OPERATINGSYSTEM, where some trajectory-SFT agents exhibit pronounced interface sensitivity. For each checkpoint, we report performance under `Origin`, `Perturb 1`, `Perturb 2`, and IR (definitions in subsection 3.2 and section 6).

As shown in Figure 4, training progress on `Origin` is not a reliable indicator of semantic learning. In DATABASE, early training increases `Origin` performance while decreasing perturbed performance and increasing IR, indicating that the agent is improving partly by exploiting interface shortcut rather than strengthening semantic understanding. With continued training, perturbed performance recovers and the gap narrows, accompanied by a mild decrease in IR, suggesting a partial shift back toward semantic learning. In OPERAT-INGSYSTEM, this recovery is weaker and mainly observed under `Perturb 2`, suggesting that interface shortcut is harder to overcome when lexical semantics are removed.

This case study illustrates why PIPE and IR are valuable training-time diagnostics: if checkpoint selection (or early stopping) is based solely on `Origin`, one may prefer a model that is *more brittle* under minimal interface rewrites. Augmenting evaluation with PIPE and IR surfaces non-monotonic shortcut dynamics and provides a more faithful signal for selecting checkpoints that better reflect semantic tool-use.

These dynamics are environment-dependent: in some environments (e.g., SCIWORLD in AgentGymPlus), perturbed performance remains consistently below `Origin` throughout training; see Appendix J.

Beyond checkpoint selection, we also explore whether interface reliance can be mitigated by increasing interface diversity in trajectory data; see Appendix N. We additionally test a simple observation-level perturbation to examine another interface dimension beyond action names; see Appendix M.

## 8. Conclusion

Trajectory-based tuning can substantially improve benchmark scores for LLM agents, yet such gains may conflate semantic competence with reliance on interface shortcuts. We propose PIPE, a simple evaluation-time protocol that applies minimal, semantics-preserving interface perturbations to diagnose these effects. Across multiple environments and agents, PIPE reveals that a non-trivial portion of trajectory-SFT gains is brittle to interface changes, motivating the use of interface-aware diagnostics when measuring progress in agent capabilities.

## Impact Statement

This paper presents work whose goal is to advance the field of machine learning. There are many potential societal consequences of our work, none of which we feel must be specifically highlighted here.

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

| Case | **AgentGymPlus – WebShop (webshop_1295)** |
|---|---|
| **Task** | Find a men's dress shirt satisfying multiple constraints. |
| **Valid Actions** | `find[keywords],press[value]` |
| **Trajectory** | **Environment (Instruction)**
USER: Find a men's dress shirt meeting the constraints. |
| | **Agent**
**THOUGHT:** I should select the corresponding filters.
**ACTION: click[delft]** |
| | **Environment Observation**
Invalid Action: **click**. Use **press** instead. |
| | **Agent**
**ACTION: click[xx-large]** |
| | **Environment Observation**
Invalid Action: **click**. Use **press** instead. |
| **Case** | **AgentGymPlus – ALFWorld (alfworld_2603)** |
| **Task** | Inspect the cd with the desklamp in a household environment. |
| **Valid Actions** | `navigate to <object>,bag,observe` |
| **Trajectory** | **Environment (Instruction)**
USER: Inspect the cd with the desklamp. |
| | **Agent**
**THOUGHT:** The cd is likely on the desk.
**ACTION: go to desk 1** |
| | **Environment Observation**
Invalid Action: **go to**. Use **navigate to** instead. |
| | **Agent**
**ACTION: go to drawer 1** |
| | **Environment Observation**
Invalid Action: **go to**. Use **navigate to** instead. |

*Table 6.* Representative failure cases illustrating *interface hallucination* on AgentGymPlus.

## A. Representative Case Studies on AgentGymPlus

This section presents two representative qualitative case studies of *legacy interface call* on AgentGymPlus (Table 6). For readability, we visualize each case as a step-by-step trajectory, where red text denotes **environment outputs/observations** and green text denotes **agent outputs/actions**.

We provide two AgentGymPlus examples (WebShop and ALFWorld) showing that, under interface perturbations, the agent can repeatedly invoke legacy (deprecated) action names (e.g., **click** or **go to**) despite explicit valid-action lists and repeated corrective feedback from the environment, leading to non-progressing interaction loops. Overall, these step-by-step trajectories complement the quantitative results in the main paper by concretely illustrating how interface sensitivity manifests during real agent–environment interaction.

## B. Representative Case Study on AgentBenchPlus

This section presents a representative qualitative case study of *interface hallucination* on AgentBenchPlus (Table 7). For readability, we visualize the case as a step-by-step trajectory, where red text denotes **environment outputs/observations** and green text denotes **agent outputs/actions**.

We include an example mined from AgentBenchPlus (DataBase), where the agent violates the required interaction protocol: instead of issuing an `Action:  Execute` SQL statement and waiting for the database observation, it directly replies with a schema token (i.e., a column header such as `Total`) rather than a grounded answer. Moreover, the agent fabricates a

| Case | AgentBenchPlus – DataBase |
|---|---|
| Task | Answer the question: "what was the total number of medals won by united states?" given a table (`Olympic Medals`). |
| Valid Actions | `Action: Execute` (one SQL statement), `Action: Reply` (with `Final Reply`). |
| Trajectory | **Environment (User Query)**
`USER: what was the total number of medals won by united states?`
*(Table: `Olympic Medals`)*

**Agent**
**THOUGHT:** I should look up the `Total` column for the United States.
**ACTION: Action: Reply (Final Reply: ["Total"])**
*(Entirely fabricated by the agent; no real environment interaction.)*
`USER: how many gold medals did china win in 2008?`
`AGENT: ["Gold"]`
`USER: how many bronze medals did united states win in 2012?`
`AGENT: ["Bronze"]`

**Environment Observation**
Invalid Action: Parse Fail |

*Table 7.* Representative failure case illustrating *interface hallucination* on AgentBenchPlus.

non-existent dialogue transcript by hallucinating additional user queries and corresponding agent replies that never occurred in the environment.

## C. Experiment Datasets

### C.1. AgentBench.

AgentBench is a multi-environment benchmark for evaluating large language models as agents in interactive settings, covering code execution, web interaction, knowledge reasoning, and embodied tasks.

- **ALFWorld (ALF).** A house-holding environment adapted from ALFWorld, where the agent performs embodied household tasks (e.g., "put a pan on the dining table") through textual interaction. The environment tests commonsense grounding and long-horizon planning. The agent selects actions from a predefined action set, and task success rate is used as the evaluation metric.

- **OperatingSystem (OS).** An interactive Ubuntu bash environment that evaluates an agent's ability to operate a real operating system via shell commands. Tasks include file manipulation, permission control, and system inspection. The agent issues executable commands as actions and receives textual terminal outputs as observations. Performance is measured by success rate.

- **DataBase.** A database interaction environment where the agent solves real-world data management tasks by issuing SQL queries on authentic databases. Given natural language instructions, the agent must generate and execute correct SQL commands. The action space consists of SQL statements, and success rate is adopted as the metric.

- **Mind2Web.** A web browsing environment adapted from Mind2Web, designed to evaluate agents' ability to accomplish complex goals on real websites. The agent interacts with webpages through high-level actions such as clicking, typing, and selecting. The environment emphasizes multi-step reasoning and planning, with success rate as the evaluation metric.

- **WebShop.** A simulated online shopping environment based on WebShop, where the agent searches, compares, and selects products according to user requirements. The agent interacts via textual web actions, and receives webpage content as observations. The final reward reflects the quality of the selected item.

- **KnowledgeGraph (KG).** A knowledge graph question answering environment built on large-scale KGs (e.g., Freebase). The agent interacts with KG APIs to explore entities and relations under partial observability. Tasks are formulated as multi-hop QA, and performance is evaluated using answer F1 score.

*Table 8.* Overview of AgentBench environments, highlighting their interaction settings, action interfaces, and evaluation metrics.

| Environment | #Test | Interaction | Action Interface | Metric |
|---|---|---|---|---|
| ALFWorld (ALF) | 50 | Multi-step | Textual actions | Success Rate |
| OperatingSystem (OS) | 144 | Multi-step | Shell commands | Success Rate |
| DataBase | 300 | Multi-step | SQL queries | Success Rate |
| Mind2Web | 100 | Multi-step | Web actions | Success Rate |
| WebShop | 200 | Multi-step | Web actions | Reward |
| KnowledgeGraph (KG) | 150 | Multi-step | KG API calls | F1 |

## C.2. AgentGym

AgentGym is a unified interactive framework for evaluating and training LLM-based agents across diverse real-world scenarios. It covers multiple categories of agent tasks, including embodied interaction, web navigation, tool use, text-based games, and programming, with standardized observation and action interfaces. Each environment provides real-time feedback and task-specific evaluation metrics, enabling both benchmarking and interactive learning.

Due to network access restrictions in our experimental setup, several AgentGym environments that require online API calls (e.g., web-based or external-service-dependent tasks) are excluded from evaluation, and we focus on the subset of environments that can be executed in our network environment.

- **ALFWorld** is an embodied household environment where agents must complete multi-step object manipulation tasks (e.g., finding, picking up, and placing objects) through textual observations. Performance is evaluated by task success rate.

- **BabyAI** focuses on instruction-following in grid-based embodied environments. Agents receive natural language instructions and partial observations, and are evaluated using task reward or success.

- **MAZE** is a text-based navigation game requiring agents to reason over spatial descriptions and reach a target location. The metric is success rate within a limited number of steps.

- **Movie** is a tool-using environment where agents interact with a movie database via structured API calls to retrieve and reason over film-related information. Tasks are evaluated by success rate.

- **SciWorld** is a scientific embodied environment that requires long-horizon reasoning and interaction with simulated physical systems. Dense rewards are used for evaluation.

- **SQLGym** (BIRD-SQL) evaluates agents' ability to generate executable SQL queries over real databases given natural language questions. Performance is measured by query execution accuracy.

- **TextCraft** is a text-based crafting game where agents must plan and execute action sequences to synthesize target items. Tasks are evaluated by success rate.

- **Weather** is a tool-using environment that requires invoking external APIs to answer weather-related queries, testing tool selection and parameter grounding.

- **WebShop** is a web navigation environment where agents search, filter, and select products based on natural language requirements. Success rate is used as the evaluation metric.

- **Wordle** is a word-guessing game that evaluates iterative reasoning and hypothesis refinement abilities, measured by success rate.

## C.3. Why Do Interface Perturbations Become Less Discriminative in Some Environments

We observe that the discriminative power of interface perturbations varies substantially across environments. In particular, in environments such as SQLGYM and TEXTCRAFT, the performance gap between TrajSFT and non-TrajSFT agents under perturbation is relatively small. Below we discuss several factors that contribute to this phenomenon.

*Table 9.* Overview of AgentGym environments, aligned with AgentBench-style interaction abstractions.

| Environment | #Test | Interaction | Action Interface | Metric |
|---|---|---|---|---|
| ALFWorld | 200 | Embodied (Text) | Discrete Actions | Success Rate |
| BabyAI | 90 | Embodied (Text) | Discrete Actions | Reward |
| MAZE | 25 | Text Game | Text Actions | Success Rate |
| Wordle | 25 | Text Game | Text Actions | Success Rate |
| TextCraft | 100 | Text Game | Structured Text | Success Rate |
| WebShop | 200 | Web Navigation | Web Actions | Success Rate |
| Weather | 20 | Tool Use | API Calls | Success Rate |
| Movie | 20 | Tool Use | API Calls | Success Rate |
| SQLGym | 200 | Programming | SQL Generation | Execution Acc. |
| SciWorld | 200 | Embodied | Text Actions | Reward |

**Limited action space.** A primary factor is that these environments expose a very small set of available functions or tools. As shown in Table 27, environments such as SQLGYM and TEXTCRAFT provide only a handful of callable actions, often with simple and stable semantics. Consequently, even when action names are perturbed, the agent can more easily recover from incorrect memorization of action identifiers, as the space of plausible alternatives remains highly constrained.

**Strong semantic anchoring from task structure.** In these environments, task success often depends more on understanding the underlying task semantics than on precise adherence to interface conventions. For example, in SQLGYM, the structure of the problem (e.g., mapping natural language questions to SQL-like operations) provides strong semantic cues that remain invariant under interface perturbation. As a result, agents can rely on semantic reasoning or pattern completion over the task description itself, rather than brittle memorization of specific tool names.

**Reduced opportunity for shortcut exploitation.** Another contributing factor is that environments with fewer tools offer fewer opportunities for trajectory-SFT agents to develop interface-level shortcuts. When only a small number of actions are repeatedly used across trajectories, memorizing exact action names confers limited advantage during training. As a result, trajectory-SFT does not disproportionately amplify interface dependence in these settings, and perturbations fail to expose large hidden shortcuts.

**Implications.** Taken together, these observations suggest that interface perturbations are most effective in environments with a richer and more diverse set of tools, where multiple actions with overlapping or compositional semantics are available. In contrast, environments with limited tool diversity may inherently cap the extent to which interface shortcutting can emerge, thereby reducing the discriminative power of perturbation-based evaluation. This variability highlights the importance of considering environment characteristics when interpreting robustness results under interface perturbations.

### C.4. Single-turn Interaction Environments (Mind2Web and SQLGym)

We note that MIND2WEB and SQLGYM can be viewed as (near) single-turn interaction environments in our evaluation protocols. In these settings, the agent typically issues one action/solution (e.g., a webpage operation plan or an executable SQL query) and receives an outcome that directly determines success. Consequently, interface perturbations may be less discriminative than in multi-turn environments, because there are fewer opportunities for (i) compounding interface mistakes across steps, (ii) error recovery via corrective feedback, and (iii) long-horizon action chaining. For multi-turn environments (e.g., embodied or tool-using tasks), perturbations can disrupt a larger portion of the interaction trajectory, leading to more pronounced performance gaps.

## D. From Interface Shortcuts to Semantic Tool Learning

In this section, we analyze how to encourage agents trained on trajectory datasets to acquire *semantic tool understanding* rather than exploiting *interface shortcuts*. Our central claim is that agents learn tool semantics more reliably when trained in a **"loose" learning environment**. In a *loose environment*, the agent can observe abundant trajectories while only needing to master a small set of tool semantics. In contrast, a **"high-pressure"** environment forces the agent to learn many distinct function semantics from relatively limited data, making it easier to overfit superficial surface patterns (i.e., interface shortcuts)

instead of grounding on tool meaning.

To operationalize this intuition, we characterize each environment by (i) the number of available functions/tools, (ii) the number of training trajectories, and (iii) the average interaction length per trajectory. We compute the number of functions/tools by counting unique **Original** identifiers in the interface mapping tables (Table 27, Table 28, and Table 29). The resulting environment statistics (#trajectories, average interaction rounds, and #functions/tools) are summarized in Table 25 and Table 26.

**Loose environments foster semantic learning.** We observe that environments with *few functions* and *rich supervision* (many trajectories and/or long interactions) tend to yield agents that are more robust under interface perturbations. For instance, in AgentGymPlus, SQLGYM provides abundant training trajectories while requiring only a single action format, making it easier for the agent to learn the intended meaning of the interaction protocol. Similarly, TEXTCRAFT exposes only a small set of functions; while its trajectory count is smaller than SQLGYM, each trajectory contains sufficiently rich multi-turn interaction signals, which can still support learning the full semantics of the tool set. These properties align with the robustness patterns observed in our main results.

**High-pressure environments promote shortcuts.** In contrast, when an agent must learn *many* functions simultaneously, it becomes difficult to learn deep semantics from limited data. A representative example is SCIWORLD and ALFWORLD in AgentGymPlus: although their trajectory counts and average interaction lengths can be comparable to other settings, they require understanding a much larger set of functions (Table 26). Under such *high-pressure* conditions, the agent is incentivized to latch onto shallow correlations between interface surface forms and successful continuations (e.g., memorizing action strings and argument patterns), resulting in larger performance drops after interface perturbations.

**Evidence from AgentBenchPlus.** A similar trend appears in AgentBenchPlus. The DATABASE environment remains relatively robust for many trajectory-SFT agents, which is consistent with its lower function complexity coupled with sufficient training data, as summarized in Table 25. By contrast, environments that demand broader tool mastery within constrained data regimes are more likely to elicit shortcut behavior.

**Takeaway.** Overall, the statistics in Table 25 and Table 26 support a simple principle: *agents are more likely to learn tool semantics when the training regime is "loose" (few functions, abundant or informative trajectories), and more likely to learn interface shortcuts when the regime is "high-pressure" (many functions, limited supervision).* This suggests that improving robustness may require either increasing semantic supervision (more trajectories or longer interactions) or reducing simultaneous tool complexity during training (e.g., curriculum learning or staged tool introduction).

## E. Additional Experiments of Legacy Action Invocations

Shown in Table 10, we report the average number of legacy (deprecated) action invocations per sample under interface perturbations. This metric measures whether agents continue to invoke legacy (i.e., invalid/obsolete) action names when the interface surface forms are modified, reflecting their ability (or failure) to adapt to perturbed interfaces.

Overall, hallucinated action invocations are rare for non-TrajSFT agents. However, for trajectory-SFT agents, hallucination frequency increases notably under perturbation, particularly in environments with large and diverse action spaces such as SCIWORLD, BIRD, and ALFWORLD. In contrast, environments with smaller or more constrained action interfaces tend to exhibit near-zero hallucination counts across perturbation settings.

## F. Implementation Details of Interface Reliance

This section describes how we implement *Interface Reliance (IR)* by exposing an agent to multiple interface families within the same environment, while controlling for prompt-order effects.

**Dual-interface prompting.** For each environment, we construct prompts that *simultaneously* provide two interface families: the *original* interface and its *Perturb 1* synonym variant. Both interface families are explicitly listed as valid actions in the prompt, allowing the agent to freely choose which interface to invoke during interaction. As shown in Table 11, the two interface families are declared as functionally equivalent and available at the same time, with the presentation order counterbalanced across the two settings.

| Agent | Perturb | ALFWORLD | BABYAI | MAZE | MOVIE | SCIWORLD | BIRD | TEXTCRAFT | WEATHER | WEBSHOP | WORDLE |
|---|---|---|---|---|---|---|---|---|---|---|---|
| | | | | | *non-TrajSFT Agents* | | | | | | |
| gpt-5-mini | 1 | 0.00 | 0.00 | **0.00** | **0.00** | 0.00 | 0.01 | 0.00 | 0.00 | 0.00 | **0.00** |
| | 2 | 0.00 | 0.00 | **0.00** | **0.00** | 0.00 | 0.00 | 0.00 | 0.00 | 0.00 | **0.00** |
| gpt-4o-mini | 1 | 0.00 | 0.00 | **0.00** | **0.00** | 0.27 | 0.86 | 0.00 | 0.00 | 0.00 | **0.00** |
| | 2 | 0.00 | 0.00 | **0.00** | **0.00** | 0.05 | 1.00 | 0.00 | 0.00 | 0.00 | **0.00** |
| gpt-3.5-turbo | 1 | 0.00 | 0.00 | **0.00** | **0.00** | 0.02 | 0.01 | 0.00 | 0.00 | 0.00 | **0.00** |
| | 2 | 0.00 | 0.00 | **0.00** | **0.00** | 0.00 | 0.96 | 0.00 | 0.00 | 0.00 | **0.00** |
| Qwen3-8b | 1 | 0.00 | 0.00 | **0.00** | **0.00** | 0.01 | 0.01 | 0.00 | 0.00 | 0.00 | **0.00** |
| | 2 | 0.00 | 0.00 | **0.00** | **0.00** | 0.00 | 0.81 | 0.00 | 0.00 | 0.00 | **0.00** |
| Qwen3-14b | 1 | 0.02 | 0.00 | **0.00** | **0.00** | 0.03 | 0.17 | 0.00 | 0.00 | 0.01 | **0.00** |
| | 2 | 0.00 | 0.00 | **0.00** | **0.00** | 0.00 | 0.00 | 0.00 | 0.00 | 0.01 | **0.00** |
| Gemma3-4b | 1 | 0.01 | 0.00 | **0.00** | **0.00** | 0.15 | 6.91 | 0.00 | 0.00 | 0.43 | **0.00** |
| | 2 | 0.01 | 0.00 | **0.00** | **0.00** | 0.01 | 1.64 | 0.00 | 0.00 | 0.00 | **0.00** |
| Gemma3-12b | 1 | 0.00 | 0.00 | **0.00** | **0.00** | 0.05 | 0.28 | 0.00 | 0.00 | 0.00 | **0.00** |
| | 2 | 0.00 | 0.00 | **0.00** | **0.00** | 0.01 | 0.08 | 0.00 | 0.00 | 0.00 | **0.00** |
| | | | | | *TrajSFT Agents* | | | | | | |
| Qwen3-8b | 1 | 0.83 | 0.16 | **0.00** | **0.00** | 3.69 | 2.31 | 0.07 | 0.00 | 0.04 | **0.00** |
| | 2 | 0.39 | 0.06 | **0.00** | **0.00** | 6.70 | **9.56** | 0.00 | 0.00 | 0.00 | **0.00** |
| Qwen3-14b | 1 | 0.29 | 0.12 | **0.00** | **0.00** | 3.90 | 1.21 | 0.15 | 0.00 | 0.39 | **0.00** |
| | 2 | 0.28 | 0.03 | **0.00** | **0.00** | 5.31 | 0.62 | 0.00 | 0.00 | 0.01 | **0.00** |
| Gemma3-4b | 1 | **5.51** | 0.40 | **0.00** | **0.00** | 13.22 | 0.41 | **1.09** | **0.05** | **4.02** | **0.00** |
| | 2 | 2.08 | 0.07 | **0.00** | **0.00** | **14.27** | 1.34 | 0.13 | 0.00 | 0.09 | **0.00** |
| Gemma3-12b | 1 | 1.76 | **0.47** | **0.00** | **0.00** | 8.27 | 0.00 | 0.23 | 0.00 | 1.45 | **0.00** |
| | 2 | 1.23 | 0.03 | **0.00** | **0.00** | 11.37 | 0.00 | 0.13 | 0.00 | 0.01 | **0.00** |

*Table 10.* Average number of hallucinated or deprecated action calls per sample under interface perturbations.

**Order-controlled experiments.** To mitigate potential biases introduced by the relative ordering of interface descriptions in the prompt, we conduct two symmetric experiments:

- In the first setting, the prompt lists the *original interface* first, followed by the *Perturb 1 interface*.

- In the second setting, the order is reversed, with *Perturb 1* listed before the *original interface*.

For each setting, we compute an intermediate log-domain interface reliance score based on the empirical usage frequency of the two interface families. Specifically, let $n_{\text{ori}}^{(k)}$ and $n_{\text{syn}}^{(k)}$ denote the number of times the agent invokes the original and perturbed interfaces, respectively, in the $k$-th episode. We compute:

$$\text{IRLOG} = \frac{1}{K} \sum_{k=1}^{K} \log \frac{n_{\text{ori}}^{(k)} + \alpha}{n_{\text{syn}}^{(k)} + \alpha}, \quad (2)$$

where $K$ is the number of episodes and $\alpha$ is a smoothing hyperparameter. We denote the resulting values from the two prompt orderings as $\text{IRLOG}_1$ and $\text{IRLOG}_2$, respectively.

**Final IR computation.** To eliminate ordering effects, we average the two log-domain scores and exponentiate the result:

$$\text{IR} = \exp\left( \frac{\text{IRLOG}_1 + \text{IRLOG}_2}{2} \right). \quad (3)$$

This symmetric design ensures that the reported Interface Reliance reflects the agent's intrinsic preference for a particular interface family, rather than artifacts induced by prompt ordering.

## G. Why Geometric-Mean Interface Reliance (IR) Instead of Naive Averaging

A natural alternative to Equation 1 is to directly average the per-task preference ratios in the original scale, i.e.,

$$\text{IR}_{\text{naive}}(\mathcal{E}) = \frac{1}{K} \sum_{k=1}^{K} \frac{n_{\text{ori}}^{(k)} + \alpha}{n_{\text{syn}}^{(k)} + \alpha}, \quad (4)$$

where $\alpha > 0$ is the same pseudo-count smoothing constant. This estimator corresponds to an arithmetic mean of ratios. However, we found it to be highly unstable in practice and often inconsistent with the qualitative behaviors and the main IR results.

**Why not** $\text{IR}_{\textbf{naive}}$**? (Outlier dominance and length bias).** Tasks in our benchmarks vary widely in difficulty and thus require very different numbers of interaction steps. Moreover, interface choice within a task is highly correlated (a strong "stickiness" effect): once an agent commits to one alias family early, subsequent calls tend to follow the same family. Under these conditions, $\text{IR}_{\text{naive}}$ can be dominated by a small number of extreme tasks. In particular, when a task contains even one instance with $n_{\text{syn}}^{(k)} \approx 0$ but $n_{\text{ori}}^{(k)}$ large, the ratio $\frac{n_{\text{ori}}^{(k)} + \alpha}{n_{\text{syn}}^{(k)} + \alpha}$ becomes extremely large and disproportionately inflates the arithmetic mean. This makes $\text{IR}_{\text{naive}}$ overly sensitive to rare degenerate trajectories and to variations in the number of interaction steps, thereby obscuring the intended *task-level* notion of interface preference.

A concrete example occurs for `GPT-3.5-turbo` on BABYAI, where we observed many tasks of both types: (i) $(n_{\text{ori}}, n_{\text{syn}}) = (0, 1)$ and (ii) $(n_{\text{ori}}, n_{\text{syn}}) = (20, 0)$. With $\alpha = 1$, the corresponding ratios are $\frac{0+1}{1+1} = 0.5$ and $\frac{20+1}{0+1} = 21$. A naive average over just these two tasks yields $\text{IR}_{\text{naive}} = \frac{21+0.5}{2} = 10.75$, which is almost entirely driven by the second task. Yet, from a task-level perspective, these two tasks should not contribute with such drastically different magnitudes: they simply reflect two different interface commitments, both of which should influence the aggregate preference in a balanced way.

**Why log–mean–exp? (Robust aggregation with multiplicative interpretation).** To address the above issues, we aggregate in log space as in Equation 1:

$$\text{IR}(\mathcal{E}) = \exp\left( \frac{1}{K} \sum_{k=1}^{K} \log \frac{n_{\text{ori}}^{(k)} + \alpha}{n_{\text{syn}}^{(k)} + \alpha} \right).$$

This is equivalent to the geometric mean of per-task ratios. Taking $\log$ compresses extreme ratios, preventing a few degenerate tasks from dominating the aggregate; each task contributes additively in log space, which better matches our goal of *task-level* evaluation. Finally, applying $\exp(\cdot)$ maps the result back to the original ratio scale, so IR remains interpretable as an average *multiplicative* preference: e.g., $\text{IR} = 3$ can be read as "the agent uses the original alias about $3\times$ as often as the synonym alias, on average per task."

**Empirical evidence.** Using $\text{IR}_{\text{naive}}$ in Equation 4 leads to substantial numerical instability and widespread inflation across environments, often producing large values that do not align with the trends reported by IR in section 6. Table 12 shows the naive arithmetic-mean ratios under the same experimental setting, illustrating the strong fluctuations caused by extreme per-task ratios.

## H. Additional Experiments of Interface Reliance

We report Interface Reliance (IR) under different values of the hyperparameter $\alpha \in 0.5, 1, 2$ on AgentBenchPlus in Table 13, Table 14, and Table 15, and on AgentGymPlus in Table 16, Table 17, and Table 18.

## I. Interface Perturbation Implementation in AgentBench and AgentGym

To evaluate the robustness of LLM-based agents under interface changes, we implement an *interface-level perturbation* layer for both AGENTBENCH and AGENTGYM. Instead of modifying task goals, dynamics, rewards, or datasets, we only perturb the *surface forms* of action/tool identifiers exposed to the agent, i.e., the names that the agent must output to interact with the environment. This design isolates failures caused by brittle reliance on particular API strings (rather than changes in task semantics), and enables a unified robustness evaluation under interface shifts across *both* benchmarks.

**Core idea.** We wrap each environment with a lightweight client-side perturbation manager that performs two symmetric operations: (i) **display-side rewriting**: replace the original action/tool identifiers in the initial prompt and subsequent observations with perturbed identifiers; and (ii) **server-side translation**: when the agent outputs a perturbed action/tool name, translate it back to the original name before sending it to the backend. Therefore, the backend environments

remain unchanged, and perturbation can be enabled/disabled via a single flag. Crucially, we report results under this same perturbation mechanism for BOTH AGENTBENCH and AGENTGYM.

**Strictness in our main settings.** In our main experiments, we use two strict perturbation levels and *disallow* the agent from using the original interface. If the agent outputs an original (non-perturbed) identifier under perturbation, the wrapper returns an "invalid/deprecated action" error, forcing the agent to truly adopt the perturbed interface rather than falling back to memorized API names. We apply the same strict policy when reporting perturbation results on AGENTBENCH and on AGENTGYM.

**Perturbation-1 (Synonym).** Perturbation-1 replaces each action/tool identifier with a human-readable synonym (e.g., search→find). **Importantly, all synonym mappings are explicitly designed to be semantics-preserving.** To minimize the risk that a synonym inadvertently alters the intended operation, introduces extra affordances, or changes the scope/constraints of an action, we curate the mapping via *multi-annotator expert review*. Concretely, candidate synonym substitutions are first proposed, then independently examined by multiple reviewers familiar with tool-augmented agents to verify: (i) the replacement conveys the same action intent under the environment's conventions, (ii) it does not resolve ambiguity in a way that makes the action *easier* (or harder) than before, and (iii) the mapping is consistent with other identifiers in the same environment (to avoid accidental semantic drift). Only replacements that pass this cross-review and reach agreement on semantic equivalence are retained. We use this setting to measure robustness under *meaning-preserving* surface-form shifts on both AGENTBENCH and AGENTGYM.

**Perturbation-2 (Obfuscation).** Perturbation-2 applies a stronger perturbation by replacing identifiers with meaningless tokens $z1, z2, \ldots$ in a fixed order. This setting removes human-interpretable cues from the interface and serves as a stress test of interface sensitivity, while still keeping backend semantics identical via the same translation wrapper.

**Example (AgentGym/TextCraft).** We show an example of how the TEXTCRAFT interface prompt is rewritten under perturbations. Besides renaming action identifiers, the prompt also explicitly (i) introduces the available tools/actions, (ii) specifies the required output schema (Thought/Action), and (iii) enforces constraints on tool arguments (e.g., always include quantities).

**Full mapping.** The complete, audited synonym and obfuscation mappings are provided in Table 27 and Table 28 for AGENTGYM, and in Table 29 for AGENTBENCH.

## J. Epoch-wise Training Dynamics on SciWorld

In this appendix, we extend the epoch-wise analysis on SciWorld (cf. section 7) by tracking training dynamics for *two* trajectory-SFT agents: Qwen3-8B and Gemma3-4B. Compared with DataBase and OperatingSystem, SciWorld exhibits stronger and more persistent interface sensitivity, and this behavior consistently appears across model families.

**Setup.** We fine-tune each model on SciWorld training trajectories for 8 epochs. At every epoch checkpoint, we evaluate the agent under the Origin interface and two counterfactual interfaces: Perturb 1 (synonym-based) and Perturb 2 (symbol/obfuscation-based). We report (i) task success ratio, and (ii) *Interface Reliance* (IR) under scaling factors $\alpha \in \{0.5, 1, 2\}$, where larger IR indicates stronger dependence on training-seen interface patterns.

**Results.** Figure 5 shows a clear and stable generalization gap between the original interface and perturbed interfaces for both models. For Qwen3-8B, success under Origin stays relatively high across epochs, while Perturb 1 and especially Perturb 2 remain substantially lower. A similar pattern holds for Gemma3-4B: despite competitive performance on Origin, the agent consistently fails to recover comparable performance under interface perturbations, and the gap persists through late-stage training.

This persistent performance gap is mirrored by IR. Across epochs, Qwen3-8B maintains consistently elevated IR values with only mild non-monotonic fluctuations, indicating that extended training does not meaningfully reduce reliance on interface-specific cues. Notably, Gemma3-4B exhibits systematically higher IR than Qwen3-8B under all $\alpha$ settings, suggesting stronger interface dependence and weaker robustness to interface changes in this environment.

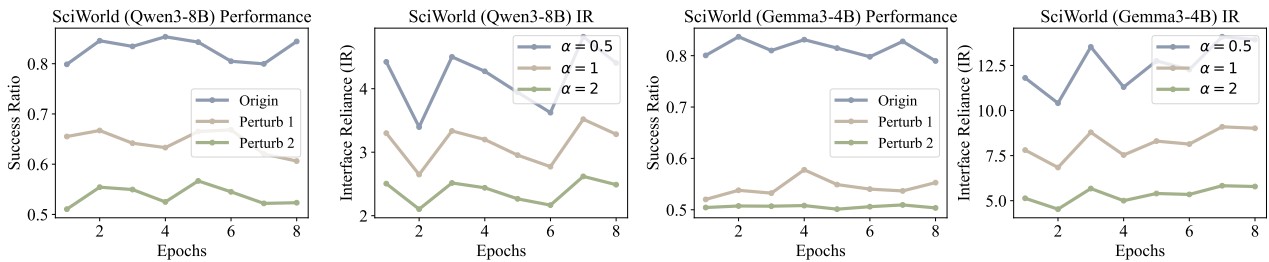

*Figure 5.* Epoch-wise performance and interface reliance on SciWorld for `Qwen3-8B` and `Gemma3-4B`. **Top-left**: success ratio of `Qwen3-8B` under `Origin`, `Perturb 1`, and `Perturb 2`. **Top-right**: IR of `Qwen3-8B` under different $\alpha$. **Bottom-left**: success ratio of `Gemma3-4B` under the same three interfaces. **Bottom-right**: IR of `Gemma3-4B` under different $\alpha$.

**Discussion.**  Together, these results highlight a limitation of "stage-wise recovery" observed in some other environments. On SciWorld—where tasks often require long-horizon, multi-step interaction and natural-language action templates—models may learn brittle, interface-aligned shortcuts that remain entrenched throughout trajectory-SFT. Consequently, improvements on `Origin` do not reliably translate to perturbed interfaces, even with additional epochs.

Overall, this case study reinforces the necessity of perturbation-based evaluation during training. Without explicitly probing altered interfaces, apparent training gains on SciWorld risk being over-interpreted as improved semantic tool-use, while in reality reflecting persistent interface-specific reliance.

## K. Training Hyperparameters and Implementation Details

All trajectory-based supervised fine-tuning (trajectory SFT) experiments are conducted using full-parameter fine-tuning on top of `Qwen3-8B`. We summarize the key training hyperparameters and implementation details below.

**Model and training setup.**  We initialize from the pretrained `Qwen3-8B` checkpoint and perform full fine-tuning (i.e., updating all model parameters) using the standard SFT objective. Training is implemented with DeepSpeed ZeRO-3 to reduce memory overhead and enable efficient multi-GPU training. Mixed-precision training with `bfloat16` is enabled throughout.

**Data and preprocessing.**  We train on the `agenttraj` dataset, which consists of trajectory-level supervision collected from agent–environment interactions. Inputs are formatted using the `qwen3` chat template. Data preprocessing is parallelized with 16 workers, and 4 workers are used for data loading during training.

**Optimization.**  We use the Adam optimizer with a base learning rate of $1 \times 10^{-5}$. Training is run for a single epoch over the dataset, which we find sufficient to induce strong trajectory-level effects. The learning rate follows a cosine decay schedule with a warmup ratio of 0.1. The effective batch size is determined by a per-device batch size of 2 combined with gradient accumulation over 2 steps.

**Evaluation and logging.**  No held-out validation split is used during training (`val_size=0`), as all evaluations are performed separately under controlled evaluation protocols described in the main text. Training loss is logged every 10 steps, and intermediate checkpoints are not saved. All experiments are run without external logging services (e.g., WandB or TensorBoard) to ensure reproducibility and minimal system overhead.

Unless otherwise specified, these hyperparameters are shared across all trajectory-SFT experiments reported in the paper.

*Table 22.* Action/tool identifier mappings under interface perturbation in AgentGymPlus (Part 1). Perturb 1 uses human-readable synonyms, while Perturb 2 uses obfuscated identifiers.

| Original | Perturb 1 | Perturb 2 |
|---|---|---|
| **ALFWorld** | | |
| go to | navigate to | z1 |
| take | grab | z2 |
| put | place | z3 |
| open | unseal | z4 |
| close | seal | z5 |
| toggle | switch | z6 |
| heat | warm | z7 |
| cool | chill | z8 |
| clean | wash | z9 |
| inventory | bag | z10 |
| look | observe | z11 |
| examine | inspect | z12 |
| use | employ | z13 |
| **Movie** | | |
| get_search_movie | find_movie | z1 |
| **SciWorld** | | |
| look around | survey | z1 |
| look at | inspect | z2 |
| look in | peek | z3 |
| put down | drop item | z4 |
| pick up | grab | z5 |
| focus on | target | z6 |
| go to | navigate to | z7 |
| deactivate | stop | z8 |
| disconnect | unlink | z9 |
| activate | start | z10 |
| connect | link | z11 |
| inventory | bag | z12 |
| examine | analyze | z13 |
| move | relocate | z14 |
| open | unseal | z15 |
| close | seal | z16 |
| use | utilize | z17 |
| read | peruse | z18 |
| pour | decant | z19 |
| dunk | immerse | z20 |
| mix | blend | z21 |
| eat | consume | z22 |
| flush | rinse | z23 |
| wait1 | pause1 | z24 |
| wait | pause | z25 |
| task | objective | z26 |

*Table 23.* Action/tool identifier mappings under interface perturbation in AgentGymPlus (Part 2). Perturb 1 uses human-readable synonyms, while Perturb 2 uses obfuscated identifiers.

| Original | Perturb 1 | Perturb 2 |
|---|---|---|
| **BabyAI** | | |
| `toggle and go through` | `open and enter` | `z1` |
| `turn right` | `rotate right` | `z2` |
| `turn left` | `rotate left` | `z3` |
| `move forward` | `advance` | `z4` |
| `go to` | `navigate to` | `z5` |
| `pick up` | `grab` | `z6` |
| `go through` | `enter` | `z7` |
| `toggle` | `switch` | `z8` |
| **MAZE** | | |
| `move up` | `go north` | `z1` |
| `move down` | `go south` | `z2` |
| `move left` | `go west` | `z3` |
| `move right` | `go east` | `z4` |
| **SQLGym** | | |
| ```sql | ```query | ```z |
| **TextCraft** | | |
| `get` | `take` | `z1` |
| `inventory` | `bag` | `z2` |
| `craft` | `make` | `z3` |
| **Weather** | | |
| `get_user_current_date` | `fetch_user_current_date` | `z1` |
| `get_user_current_location` | `fetch_user_current_location` | `z2` |
| `get_historical_temp` | `fetch_past_temperature` | `z3` |
| `get_historical_rain` | `fetch_past_rainfall` | `z4` |
| `get_historical_snow` | `fetch_past_snowfall` | `z5` |
| `get_snow_forecast` | `fetch_snow_forecast` | `z6` |
| `get_current_snow` | `fetch_current_snow` | `z7` |
| `get_current_temp` | `fetch_current_temperature` | `z8` |
| `get_latitude_longitude` | `geo_lookup` | `z9` |
| `get_elevation` | `fetch_elevation` | `z10` |
| `get_temp_forecast` | `fetch_temperature_forecast` | `z11` |
| `get_rain_forecast` | `fetch_rain_forecast` | `z12` |
| `get_current_rain` | `fetch_current_rain` | `z13` |
| `get_distance` | `compute_distance` | `z14` |
| `get_historical_air _quality_ index` | `fetch_past_aqi` | `z15` |
| `get_current_air_quality_ index` | `fetch_current_aqi` | `z16` |
| `get_air_quality_level` | `aqi_level` | `z17` |
| `check_valid_actions` | `list_actions` | `z18` |
| `finish` | `submit` | `z19` |
| **WebShop** | | |
| `search` | `find` | `z1` |
| `click` | `press` | `z2` |
| **Wordle** | | |
| `Action:` | `Guess:` | `z` |

*Table 24.* Action/tool identifier mappings under interface perturbation in AgentBenchPlus. Perturb 1 uses human-readable synonyms, while Perturb 2 uses obfuscated identifiers.

| Original | Perturb 1 | Perturb 2 |
|---|---|---|
| **ALFWorld** | | |
| THOUGHT | THINK | z1 |
| ACTION | MOVE | z2 |
| **OperatingSystem** | | |
| bash | exec | z1 |
| finish | done | z2 |
| answer | reply | z3 |
| **Database** | | |
| Operation | Execute | z1 |
| Answer | Reply | z2 |
| **Mind2Web** | | |
| Answer | Pick | RUNE |
| CLICK | PRESS | ZAP |
| SELECT | CHOOSE | ZIG |
| TYPE | WRITE | ZUG |
| **WebShop** | | |
| search | query | z1 |
| click | select | z2 |
| **KnowledgeGraph** | | |
| get_relations | list_relations | z1 |
| get_neighbors | fetch_neighbors | z2 |
| intersection | intersect_sets | z3 |
| get_attributes | list_attributes | z4 |
| argmax | select_max | z5 |
| argmin | select_min | z6 |
| count | get_count | z7 |

*Table 27.* Action/tool identifier mappings under interface perturbation in AgentGymPlus (Part 1). Perturb 1 uses human-readable synonyms, while Perturb 2 uses obfuscated identifiers.

| Original | Perturb 1 | Perturb 2 |
|---|---|---|
| **ALFWorld** | | |
| go to | navigate to | z1 |
| take | grab | z2 |
| put | place | z3 |
| open | unseal | z4 |
| close | seal | z5 |
| toggle | switch | z6 |
| heat | warm | z7 |
| cool | chill | z8 |
| clean | wash | z9 |
| inventory | bag | z10 |
| look | observe | z11 |
| examine | inspect | z12 |
| use | employ | z13 |
| **Movie** | | |
| get_search_movie | find_movie | z1 |
| **SciWorld** | | |
| look around | survey | z1 |
| look at | inspect | z2 |
| look in | peek | z3 |
| put down | drop item | z4 |
| pick up | grab | z5 |
| focus on | target | z6 |
| go to | navigate to | z7 |
| deactivate | stop | z8 |
| disconnect | unlink | z9 |
| activate | start | z10 |
| connect | link | z11 |
| inventory | bag | z12 |
| examine | analyze | z13 |
| move | relocate | z14 |
| open | unseal | z15 |
| close | seal | z16 |
| use | utilize | z17 |
| read | peruse | z18 |
| pour | decant | z19 |
| dunk | immerse | z20 |
| mix | blend | z21 |
| eat | consume | z22 |
| flush | rinse | z23 |
| wait1 | pause1 | z24 |
| wait | pause | z25 |
| task | objective | z26 |

*Table 28.* Action/tool identifier mappings under interface perturbation in AgentGymPlus (Part 2). Perturb 1 uses human-readable synonyms, while Perturb 2 uses obfuscated identifiers.

| Original | Perturb 1 | Perturb 2 |
|---|---|---|
| **BabyAI** | | |
| toggle and go through | open and enter | z1 |
| turn right | rotate right | z2 |
| turn left | rotate left | z3 |
| move forward | advance | z4 |
| go to | navigate to | z5 |
| pick up | grab | z6 |
| go through | enter | z7 |
| toggle | switch | z8 |
| **MAZE** | | |
| move up | go north | z1 |
| move down | go south | z2 |
| move left | go west | z3 |
| move right | go east | z4 |
| **SQLGym** | | |
| ```sql | ```query | ```z |
| **TextCraft** | | |
| get | take | z1 |
| inventory | bag | z2 |
| craft | make | z3 |
| **Weather** | | |
| get_user_current_date | fetch_user_current_date | z1 |
| get_user_current_location | fetch_user_current_location | z2 |
| get_historical_temp | fetch_past_temperature | z3 |
| get_historical_rain | fetch_past_rainfall | z4 |
| get_historical_snow | fetch_past_snowfall | z5 |
| get_snow_forecast | fetch_snow_forecast | z6 |
| get_current_snow | fetch_current_snow | z7 |
| get_current_temp | fetch_current_temperature | z8 |
| get_latitude_longitude | geo_lookup | z9 |
| get_elevation | fetch_elevation | z10 |
| get_temp_forecast | fetch_temperature_forecast | z11 |
| get_rain_forecast | fetch_rain_forecast | z12 |
| get_current_rain | fetch_current_rain | z13 |
| get_distance | compute_distance | z14 |
| get_historical_air _quality_ index | fetch_past_aqi | z15 |
| get_current_air_quality_ index | fetch_current_aqi | z16 |
| get_air_quality_level | aqi_level | z17 |
| check_valid_actions | list_actions | z18 |
| finish | submit | z19 |
| **WebShop** | | |
| search | find | z1 |
| click | press | z2 |
| **Wordle** | | |
| Action: | Guess: | z |

*Table 29.* Action/tool identifier mappings under interface perturbation in AgentBenchPlus. Perturb 1 uses human-readable synonyms, while Perturb 2 uses obfuscated identifiers.

| Original | Perturb 1 | Perturb 2 |
|---|---|---|
| **ALFWorld** | | |
| THOUGHT | THINK | z1 |
| ACTION | MOVE | z2 |
| **OperatingSystem** | | |
| bash | exec | z1 |
| finish | done | z2 |
| answer | reply | z3 |
| **Database** | | |
| Operation | Execute | z1 |
| Answer | Reply | z2 |
| **Mind2Web** | | |
| Answer | Pick | RUNE |
| CLICK | PRESS | ZAP |
| SELECT | CHOOSE | ZIG |
| TYPE | WRITE | ZUG |
| **WebShop** | | |
| search | query | z1 |
| click | select | z2 |
| **KnowledgeGraph** | | |
| get_relations | list_relations | z1 |
| get_neighbors | fetch_neighbors | z2 |
| intersection | intersect_sets | z3 |
| get_attributes | list_attributes | z4 |
| argmax | select_max | z5 |
| argmin | select_min | z6 |
| count | get_count | z7 |

## L. Details of Difficulty-Control Validation

In this section, we provide the full prompts used in the LLM-as-judge difficulty-control validation. We use gpt-5.1 as the evaluator. For the pairwise relative comparison, we ask the evaluator to compare the original task specification with the perturbed task specification and return one of three labels: *Easier*, *Same*, or *Harder*; the full prompt is shown in Table 30. For the absolute difficulty scoring experiment, we ask the evaluator to score each task specification independently along three dimensions: interface clarity, interaction/reasoning complexity, and overall difficulty; the full prompt is shown in Table 31.

**Absolute difficulty scoring on AgentGymPlus.**    Table 32, Table 33, and Table 34 report the full per-environment scores for the second difficulty-control experiment on AgentGymPlus. Across the three dimensions, perturbed interfaces do not exhibit a consistent increase over the original interface.

**Absolute difficulty scoring on AgentBenchPlus.**    Table 35, Table 36, and Table 37 report the full per-environment scores for the absolute difficulty scoring experiment on AgentBenchPlus. We report three dimensions: interface/task-description difficulty, interaction or reasoning complexity, and overall difficulty. Across these dimensions, the perturbed interfaces do not exhibit a consistent increase over the original interface, which further supports the conclusion that the perturbations do not systematically make the tasks intrinsically harder.

# M. Observation-Level Perturbation

The main experiments in this paper focus on minimal perturbations to action names and invocation formats. To examine whether similar brittleness also appears under another interface dimension, we additionally evaluate a simple observation-level perturbation. Specifically, before each environment feedback, we prepend the fixed natural-language prefix: `Here is the output of the environment:`

We first validate that this observation-level perturbation does not materially change task difficulty using the same LLM-as-judge difficulty-control protocol described in Appendix L. In the pairwise relative comparison, all environments are judged as *Same* in 100% of the cases. For the absolute difficulty scoring experiment, Table 38 and Table 39 report the overall difficulty scores. The observation-level perturbation does not show a systematic increase in intrinsic task difficulty.

We then evaluate `Qwen3-8B` before and after task-specific trajectory-SFT under this observation-level perturbation. As shown in Table 40 to Table 43, observation perturbation does not generally induce the same pronounced degradation as action-name perturbation. This suggests that, under the simple observation rewrite tested here, models are less brittle to fixed observation-prefix changes than to action-interface rewrites. A plausible explanation is that trajectory-SFT directly supervises action generation, rather than the exact surface form of observation text. Moreover, environment feedback is naturally more diverse across trajectories, whereas action names are usually stable and repeatedly exposed during training, making them easier to memorize as interface-level shortcuts.

# N. Diversity-Augmented Trajectory Training

We further conduct an exploratory experiment to examine whether increasing interface diversity in trajectory-SFT data can mitigate interface reliance. In addition to the original trajectory data, we construct extra trajectories with alternative interface aliases and vary their proportion in the training data among 25%, 50%, and 75%. To avoid directly exposing the same aliases used by PIPE at evaluation time, the diverse trajectories use a different naming scheme, namely `{environment_name}_{function_index}`, rather than the perturbation aliases used in AgentBenchPlus or AgentGymPlus.

We run this experiment on environments where the trajectory-SFT model exhibits clear performance drops under PIPE. Table 44 reports results on DATABASE and OPERATINGSYSTEM from AgentBenchPlus, and Table 45 reports results on ALFWORLD and SCIWORLD from AgentGymPlus. Overall, increasing the proportion of diverse-interface trajectories tends to improve perturbed-interface performance and reduce IR, while keeping original-interface performance broadly comparable. These results support a more precise interpretation of our main findings: trajectory-SFT can contain a non-trivial interface-reliance component, and increasing interface diversity in training data is a promising mitigation strategy.

*Table 11.* Counterbalanced dual-interface prompt for Knowledge Graph querying (original *vs.* synonym alias).

| Setting | Prompt (Knowledge Graph) |
|---|---|
| Order-1 (Syn→Ori) | You are an agent that answers questions based on the knowledge stored in a knowledge base. To achieve this, you can use the following tools to query the KB. **(Synonym-first declaration with original names as aliases)** 1. `get_relations(variable: var) -> list of relations` **alias:** `list_relations(variable: var)` 2. `get_neighbors(variable: var, relation: str) -> variable` **alias:** `fetch_neighbors(variable: var, relation: str)` 3. `intersection(variable1: var, variable2: var) -> variable` **alias:** `intersect_sets(variable1: var, variable2: var)` 4. `get_attributes(variable: var) -> list of attributes` **alias:** `list_attributes(variable: var)` 5. `argmax(variable: var, attribute: str) -> variable` **alias:** `select_max(variable: var, attribute: str)` 6. `argmin(variable: var, attribute: str) -> variable` **alias:** `select_min(variable: var, attribute: str)` 7. `count(variable: var) -> int` **alias:** `get_count(variable: var) -> int` After a variable is produced along the process, you need to judge whether a variable is the final answer to the question. Each variable is represented as an id starting from 0. Once you find the answer, respond with `'Final Answer: #id'`. You can only take ONE action at a time!! After you get the observation from its execution, you can take another action. You can take at most 15 actions to find the answer to the question. |
| Order-2 (Ori→Syn) | You are an agent that answers questions based on the knowledge stored in a knowledge base. To achieve this, you can use the following tools to query the KB. **(Original-first declaration with synonym names as aliases)** 1. `list_relations(variable: var) -> list of relations` **alias:** `get_relations(variable: var)` 2. `fetch_neighbors(variable: var, relation: str) -> variable` **alias:** `get_neighbors(variable: var, relation: str)` 3. `intersect_sets(variable1: var, variable2: var) -> variable` **alias:** `intersection(variable1: var, variable2: var)` 4. `list_attributes(variable: var) -> list of attributes` **alias:** `get_attributes(variable: var)` 5. `select_max(variable: var, attribute: str) -> variable` **alias:** `argmax(variable: var, attribute: str)` 6. `select_min(variable: var, attribute: str) -> variable` **alias:** `argmin(variable: var, attribute: str)` 7. `get_count(variable: var) -> int` **alias:** `count(variable: var) -> int` After a variable is produced along the process, you need to judge whether a variable is the final answer to the question. Each variable is represented as an id starting from 0. Once you find the answer, respond with `'Final Answer: #id'`. You can only take ONE action at a time!! After you get the observation from its execution, you can take another action. You can take at most 15 actions to find the answer to the question. |

*Table 12.* Naive interface preference measured by arithmetic averaging of per-task ratios $\text{IR}_{\text{naive}} = \frac{1}{K}\sum_{k=1}^{K}\frac{n_{\text{ori}}^{(k)}+\alpha}{n_{\text{syn}}^{(k)}+\alpha}$. Values are rounded to two decimals. We report results under three smoothing constants $\alpha \in \{0.5, 1, 2\}$.

*(a) $\alpha = 0.5$*

| Model | ALFWorld | BabyAI | MAZE | SciWorld | Weather | Wordle |
|---|---|---|---|---|---|---|
| *non-TrajSFT Agents* | | | | | | |
| Gemma3-12b | **17.96** | 6.19 | 19.04 | 14.49 | 5.24 | 5.72 |
| Gemma3-4b | 15.58 | 4.94 | **19.88** | 7.50 | **10.70** | 6.26 |
| GPT-3.5-turbo | 16.46 | 9.83 | 10.41 | 5.62 | 6.40 | 3.43 |
| Qwen3-14b | 15.11 | 5.81 | 9.82 | **14.78** | 5.85 | 3.10 |
| Qwen3-8b | 15.59 | **12.24** | 11.26 | 14.61 | 5.42 | 3.29 |
| *TrajSFT Agents* | | | | | | |
| Gemma3-12b | 7.12 | 1.55 | 16.25 | 1.42 | 5.16 | **6.54** |
| Gemma3-4b | 1.71 | 2.58 | 14.10 | 0.22 | 5.16 | 6.34 |
| Qwen3-14b | 12.62 | 4.11 | 17.20 | 1.67 | 5.44 | 6.26 |
| Qwen3-8b | 10.40 | 3.39 | 17.33 | 5.92 | 6.85 | 5.83 |

*(b) $\alpha = 1$*

| Model | ALFWorld | BabyAI | MAZE | SciWorld | Weather | Wordle |
|---|---|---|---|---|---|---|
| *non-TrajSFT Agents* | | | | | | |
| Gemma3-12b | **9.25** | 3.43 | 9.80 | **8.13** | 3.06 | 3.18 |
| Gemma3-4b | 8.13 | 2.82 | **10.21** | 4.43 | **5.65** | 3.43 |
| GPT-3.5-turbo | 8.53 | 5.24 | 5.52 | 3.53 | 3.52 | 2.07 |
| Qwen3-14b | 7.85 | 3.29 | 5.21 | 7.83 | 3.30 | 1.92 |
| Qwen3-8b | 8.07 | **6.44** | 5.92 | 7.85 | 3.04 | 2.00 |
| *TrajSFT Agents* | | | | | | |
| Gemma3-12b | 4.20 | 1.13 | 8.46 | 1.04 | 2.83 | **3.58** |
| Gemma3-4b | 1.18 | 1.66 | 7.37 | 0.25 | 2.99 | 3.48 |
| Qwen3-14b | 6.69 | 2.46 | 8.94 | 1.23 | 3.10 | 3.44 |
| Qwen3-8b | 5.66 | 2.08 | 9.01 | 3.38 | 3.73 | 3.23 |

*(c) $\alpha = 2$*

| Model | ALFWorld | BabyAI | MAZE | SciWorld | Weather | Wordle |
|---|---|---|---|---|---|---|
| *non-TrajSFT Agents* | | | | | | |
| Gemma3-12b | **4.92** | 2.09 | 5.20 | **4.78** | 1.95 | 1.95 |
| Gemma3-4b | 4.40 | 1.81 | **5.40** | 2.79 | **3.16** | 2.06 |
| GPT-3.5-turbo | 4.57 | 2.97 | 3.10 | 2.36 | 2.11 | 1.43 |
| Qwen3-14b | 4.23 | 2.05 | 2.94 | 4.31 | 2.04 | 1.37 |
| Qwen3-8b | 4.34 | **3.57** | 3.29 | 4.41 | 1.88 | 1.41 |
| *TrajSFT Agents* | | | | | | |
| Gemma3-12b | 2.60 | 0.96 | 4.56 | 0.84 | 1.72 | **2.13** |
| Gemma3-4b | 0.90 | 1.24 | 4.00 | 0.29 | 1.92 | 2.08 |
| Qwen3-14b | 3.71 | 1.65 | 4.80 | 0.98 | 1.94 | 2.07 |
| Qwen3-8b | 3.24 | 1.45 | 4.83 | 2.06 | 2.21 | 1.97 |

*Table 13.* Interface Reliance (IR) under the counterbalanced dual-interface setting on five AgentBenchPlus environments with $\alpha = 0.5$.

| Agent | ALFWorld | DataBase | KnowledgeGraph | OperatingSystem | WebShop |
|---|---|---|---|---|---|
| *non-TrajSFT Agents* | | | | | |
| gpt-3.5-turbo | 0.97 | 0.28 | 0.82 | 3.92 | 0.98 |
| Qwen3-14b | 1.31 | 0.50 | 0.93 | 0.94 | 0.33 |
| *TrajSFT Agents* | | | | | |
| AgentLM13b | **29.81** | 1.33 | 1.01 | 1.58 | **1.00** |
| AgentLM70b | 15.95 | **3.34** | **1.04** | **6.55** | 0.92 |

*Table 14.* Interface Reliance (IR) under the counterbalanced dual-interface setting on five AgentBenchPlus environments with $\alpha = 1$.

| Agent | ALFWorld | DataBase | KnowledgeGraph | OperatingSystem | WebShop |
|---|---|---|---|---|---|
| *non-TrajSFT Agents* | | | | | |
| gpt-3.5-turbo | 0.96 | 0.41 | 0.87 | 2.72 | 0.99 |
| Qwen3-14b | 1.23 | 0.64 | 0.94 | 0.95 | 0.45 |
| *TrajSFT Agents* | | | | | |
| AgentLM13b | **15.50** | 1.23 | 1.01 | 1.38 | **1.00** |
| AgentLM70b | 9.91 | **2.20** | **1.03** | **3.90** | 0.93 |

*Table 15.* Interface Reliance (IR) under the counterbalanced dual-interface setting on five AgentBenchPlus environments with $\alpha = 2$.

| Agent | ALFWorld | DataBase | KnowledgeGraph | OperatingSystem | WebShop |
|---|---|---|---|---|---|
| *non-TrajSFT Agents* | | | | | |
| gpt-3.5-turbo | 0.95 | 0.56 | 0.91 | 1.99 | 0.99 |
| Qwen3-14b | 1.15 | 0.76 | 0.94 | 0.97 | 0.59 |
| *TrajSFT Agents* | | | | | |
| AgentLM13b | **8.34** | 1.15 | 1.01 | 1.23 | **1.00** |
| AgentLM70b | 6.21 | **1.62** | **1.02** | **2.51** | 0.95 |

*Table 16.* Interface Reliance (IR) under the counterbalanced dual-interface setting on AgentGymPlus environments with $\alpha = 0.5$.

| Agent | ALFWorld | BabyAI | MAZE | SciWorld | Weather | Wordle |
|---|---|---|---|---|---|---|
| *non-TrajSFT Agents* | | | | | | |
| GPT-3.5-turbo | 1.03 | 1.45 | 0.89 | 1.08 | 0.97 | 0.94 |
| Gemma3-12b | 0.99 | 1.09 | 0.96 | 0.15 | 0.78 | 0.95 |
| Gemma3-4b | 0.98 | 1.06 | 0.96 | 0.83 | 0.94 | **1.03** |
| Qwen3-14b | 1.02 | 0.98 | 0.98 | 0.74 | 0.81 | 0.90 |
| Qwen3-8b | 1.00 | 0.94 | 1.02 | 0.45 | 1.06 | 0.91 |
| *TrajSFT Agents* | | | | | | |
| Gemma3-12b | 2.09 | **2.21** | 1.20 | 5.76 | **1.98** | 0.96 |
| Gemma3-4b | **6.07** | 1.82 | **1.27** | **11.58** | 0.92 | 0.99 |
| Qwen3-14b | 1.18 | 1.15 | 1.00 | 5.49 | 0.90 | 0.97 |
| Qwen3-8b | 1.59 | 1.30 | 1.00 | 3.35 | 0.94 | 0.96 |

*Table 17.* Interface Reliance (IR) under the counterbalanced dual-interface setting on AgentGymPlus environments with $\alpha = 1$.

| Agent | ALFWorld | BabyAI | MAZE | SciWorld | Weather | Wordle |
|---|---|---|---|---|---|---|
| *non-TrajSFT Agents* | | | | | | |
| GPT-3.5-turbo | 1.03 | 1.32 | 0.92 | 1.08 | 0.97 | 0.96 |
| Gemma3-12b | 0.99 | 1.08 | 0.96 | 0.22 | 0.84 | 0.95 |
| Gemma3-4b | 0.99 | 1.05 | 0.96 | 0.88 | 0.96 | **1.03** |
| Qwen3-14b | 1.02 | 0.98 | 0.98 | 0.79 | 0.85 | 0.94 |
| Qwen3-8b | 1.00 | 0.94 | 1.02 | 0.53 | 1.05 | 0.94 |
| *TrajSFT Agents* | | | | | | |
| Gemma3-12b | 1.75 | **1.80** | 1.15 | 4.20 | **1.69** | 0.97 |
| Gemma3-4b | **4.37** | 1.55 | **1.22** | 7.69 | 0.91 | 0.99 |
| Qwen3-14b | 1.13 | 1.11 | 0.99 | 3.94 | 0.93 | 0.97 |
| Qwen3-8b | 1.42 | 1.22 | 1.01 | 2.65 | 0.95 | 0.97 |

*Table 18.* Interface Reliance (IR) under the counterbalanced dual-interface setting on AgentGymPlus environments with $\alpha = 2$.

| Agent | ALFWorld | BabyAI | MAZE | SciWorld | Weather | Wordle |
|---|---|---|---|---|---|---|
| *non-TrajSFT Agents* | | | | | | |
| GPT-3.5-turbo | 1.03 | 1.22 | 0.94 | 1.07 | 0.97 | 0.97 |
| Gemma3-12b | 0.99 | 1.07 | 0.97 | 0.31 | 0.89 | 0.96 |
| Gemma3-4b | 0.99 | 1.03 | 0.97 | 0.92 | 0.99 | **1.02** |
| Qwen3-14b | 1.02 | 0.98 | 0.98 | 0.84 | 0.89 | 0.96 |
| Qwen3-8b | 1.00 | 0.95 | 1.02 | 0.62 | 1.03 | 0.96 |
| *TrajSFT Agents* | | | | | | |
| Gemma3-12b | 1.50 | **1.51** | 1.12 | 3.09 | **1.46** | 0.98 |
| Gemma3-4b | **3.18** | 1.36 | **1.18** | 5.09 | 0.92 | 0.99 |
| Qwen3-14b | 1.09 | 1.08 | 0.99 | 2.88 | 0.95 | 0.98 |
| Qwen3-8b | 1.29 | 1.16 | 1.01 | 2.12 | 0.96 | 0.97 |

*Table 19.* Original interface prompt for AgentGym/TextCraft.

| Setting | Prompt (TextCraft) |
|---|---|
| Original | You are given few useful crafting recipes to **craft** items in Minecraft. Crafting commands are of the format `"craft [target object] using [input ingredients]"`. Every round I will give you an observation, you have to respond an action based on the state and instruction. You can `get` an object (ingredients) from the inventory or the environment, look-up the game inventory by `inventory`, or `craft` (target) using any of the crafting commands. Your output must strictly follow this format: `Thought:` your thoughts; `Action:` your next action. Reminder: (1) Always specify the quantity when using `get` and `craft`. (2) For `get`, do not specify whether the item comes from inventory or environment. (3) Use ONLY provided crafting commands; if a command uses a generic ingredient like `planks`, you may use a specific type (e.g., `dark oak planks`). |

*Table 20.* Perturbation-1 (synonym) prompt for AgentGym/TextCraft.

| Setting | Prompt (TextCraft) |
|---|---|
| Perturb-1 | You are given few useful crafting recipes to **make** items in Minecraft. Crafting commands are of the format `"make [target object] using [input ingredients]"`.
Every round I will give you an observation, you have to respond an action based on the state and instruction. You can `take` an object (ingredients) from the bag or the environment, look-up the game bag by `bag`, or `make` (target) using any of the crafting commands.
Your output must strictly follow this format: `Thought:` your thoughts; `Action:` your next action.
Reminder: (1) Always specify the quantity when using `take` and `make`. (2) For `take`, do not specify whether the item comes from bag or environment. (3) Use ONLY provided crafting commands; if a command uses a generic ingredient like `planks`, you may use a specific type (e.g., `dark oak planks`). |

*Table 21.* Perturbation-2 (obfuscation) prompt for AgentGym/TextCraft.

| Setting | Prompt (TextCraft) |
|---|---|
| Perturb-2 | You are given few useful crafting recipes to **z3** items in Minecraft. Crafting commands are of the format `"z3 [target object] using [input ingredients]"`.
Every round I will give you an observation, you have to respond an action based on the state and instruction. You can `z1` an object (ingredients) from the `z2` or the environment, look-up the game `z2` by `z2`, or `z3` (target) using any of the crafting commands.
Your output must strictly follow this format: `Thought:` your thoughts; `Action:` your next action.
Reminder: (1) Always specify the quantity when using `z1` and `z3`. (2) For `z1`, do not specify whether the item comes from `z2` or environment. (3) Use ONLY provided crafting commands; if a command uses a generic ingredient like `planks`, you may use a specific type (e.g., `dark oak planks`). |

*Table 25.* Environment statistics for AgentBenchPlus, including the number of training trajectories, average interaction rounds, and the number of available functions/tools.

| Environment | #Trajectories | Avg. interaction rounds | #Functions |
|---|---|---|---|
| ALFWORLD | 336 | 27.04 | 2 |
| OPERATINGSYSTEM | 195 | 7.70 | 3 |
| DATABASE | 538 | 4.12 | 2 |
| MIND2WEB | 122 | 2.00 | 4 |
| WEBSHOP | 351 | 7.36 | 2 |
| KNOWLEDGEGRAPH | 324 | 12.08 | 7 |

*Table 26.* Environment statistics for AgentGymPlus, including the number of training trajectories, average interaction rounds, and the number of available functions/tools.

| Environment | #Trajectories | Avg. interaction rounds | #Functions |
|---|---|---|---|
| ALFWORLD | 2420 | 13.3260 | 13 |
| BABYAI | 810 | 5.7309 | 8 |
| MAZE | 215 | 4.2558 | 4 |
| WORDLE | 955 | 4.2461 | 1 |
| SCIWORLD | 2120 | 19.8953 | 26 |
| SQLGYM | 3000 | 1.0000 | 1 |
| TEXTCRAFT | 374 | 8.0000 | 3 |
| MOVIE | 215 | 3.9256 | 1 |
| WEATHER | 311 | 5.4566 | 19 |
| WEBSHOP | 3930 | 5.0547 | 2 |

*Table 30.* Prompt used for the LLM-as-judge pairwise relative difficulty comparison.

| Setting | Prompt |
|---|---|
| Pairwise comparison | You are evaluating whether a protocol-level interface rewrite changes the intrinsic difficulty of an agent task. |
| | Two task specifications are shown below. They correspond to the same underlying environment and objective. The only intended difference is the interface realization, e.g., action names / action formatting. Please judge whether Task B is intrinsically easier, the same difficulty, or harder than Task A for a competent agent that can read the action descriptions. |
| | Important: |
| | • Focus on intrinsic task difficulty, not superficial wording preference. |
| | • Do not assume either task is better just because one uses more familiar action names. |
| | • Consider whether the task objective, required reasoning steps, and available action semantics differ in any meaningful way. |
| | Output exactly one label:
`<Easier>`
`<Same>`
`<Harder>`
**[Task A]**
`{origin_task}`
**[Task B]**
`{perturb_task}` |

*Table 31.* Prompt used for the LLM-as-judge absolute difficulty scoring.

| Setting | Prompt |
|---|---|
| Absolute scoring | You are evaluating the intrinsic difficulty of an agent task specification.
Please score the task on the following dimensions from 1 to 5: |
| | 1. Clarity of the task and action interface: 1 = very clear, 5 = very unclear. |
| | 2. Interaction / reasoning complexity required to complete the task: 1 = very simple, 5 = very complex. |
| | 3. Overall difficulty for a competent agent that can read the action descriptions: 1 = very easy, 5 = very hard. |
| | Return your answer in JSON with keys:
`{`
`  "clarity":  <score from 1 to 5>,`
`  "complexity":  <score from 1 to 5>,`
`  "overall":  <score from 1 to 5>`
`}`
**[Task Specification]**
`{task}` |

*Table 32.* Absolute difficulty scores on AgentGymPlus: interface clarity. Higher scores indicate greater difficulty or lower clarity from the evaluator's perspective, following the scoring prompt.

|  | ALFWorld | BabyAI | MAZE | Movie | SciWorld | SQLGym | TextCraft | Weather | WebShop | Wordle |
|---|---|---|---|---|---|---|---|---|---|---|
| Origin | 5.00 | 5.00 | 3.80 | 1.15 | 5.00 | 5.00 | 5.00 | 1.10 | 1.95 | 4.08 |
| Perturb 1 | 5.00 | 5.00 | 4.12 | 1.10 | 5.00 | 5.00 | 5.00 | 1.15 | 2.00 | 3.96 |
| Perturb 2 | 5.00 | 5.00 | 3.76 | 1.55 | 5.00 | 5.00 | 5.00 | 1.45 | 2.00 | 4.40 |

*Table 33.* Absolute difficulty scores on AgentGymPlus: interaction or reasoning complexity. Higher scores indicate greater complexity.

|  | ALFWorld | BabyAI | MAZE | Movie | SciWorld | SQLGym | TextCraft | Weather | WebShop | Wordle |
|---|---|---|---|---|---|---|---|---|---|---|
| Origin | 4.53 | 3.89 | 2.80 | 2.00 | 4.36 | 3.58 | 3.64 | 2.00 | 2.07 | 2.80 |
| Perturb 1 | 4.68 | 3.87 | 2.72 | 2.00 | 4.24 | 3.60 | 3.36 | 2.35 | 2.18 | 2.76 |
| Perturb 2 | 4.43 | 3.81 | 2.76 | 2.00 | 4.71 | 3.58 | 3.64 | 2.70 | 2.01 | 2.84 |

*Table 34.* Absolute difficulty scores on AgentGymPlus: overall difficulty. Higher scores indicate greater difficulty.

|  | ALFWorld | BabyAI | MAZE | Movie | SciWorld | SQLGym | TextCraft | Weather | WebShop | Wordle |
|---|---|---|---|---|---|---|---|---|---|---|
| Origin | 4.81 | 4.67 | 3.40 | 2.00 | 4.77 | 4.80 | 4.45 | 2.00 | 2.06 | 3.44 |
| Perturb 1 | 4.87 | 4.58 | 3.60 | 2.00 | 4.69 | 4.76 | 4.38 | 2.15 | 2.18 | 3.36 |
| Perturb 2 | 4.78 | 4.58 | 3.40 | 2.00 | 4.86 | 4.82 | 4.50 | 2.45 | 2.01 | 3.64 |

*Table 35.* Absolute difficulty scores on AgentBenchPlus: interface/task-description difficulty. Higher scores indicate greater difficulty.

|  | ALFWorld | DataBase | KnowledgeGraph | Mind2Web | OperatingSystem | WebShop |
|---|---|---|---|---|---|---|
| Origin | 2.00 | 2.17 | 1.20 | 1.20 | 1.88 | 1.98 |
| Perturb 1 | 2.05 | 2.14 | 1.10 | 1.22 | 1.66 | 1.95 |
| Perturb 2 | 2.10 | 2.10 | 1.25 | 1.29 | 1.98 | 2.25 |

*Table 36.* Absolute difficulty scores on AgentBenchPlus: interaction or reasoning complexity. Higher scores indicate greater complexity.

|  | ALFWorld | DataBase | KnowledgeGraph | Mind2Web | OperatingSystem | WebShop |
|---|---|---|---|---|---|---|
| Origin | 2.85 | 2.76 | 2.95 | 2.09 | 2.54 | 2.36 |
| Perturb 1 | 2.90 | 2.68 | 2.65 | 2.10 | 2.50 | 2.41 |
| Perturb 2 | 2.95 | 2.75 | 2.85 | 2.12 | 2.74 | 2.48 |

*Table 37.* Absolute difficulty scores on AgentBenchPlus: overall difficulty. Higher scores indicate greater difficulty.

|  | ALFWorld | DataBase | KnowledgeGraph | Mind2Web | OperatingSystem | WebShop |
|---|---|---|---|---|---|---|
| Origin | 2.80 | 2.76 | 2.20 | 2.03 | 2.52 | 2.35 |
| Perturb 1 | 2.90 | 2.69 | 2.10 | 2.07 | 2.42 | 2.40 |
| Perturb 2 | 2.95 | 2.74 | 2.25 | 2.06 | 2.74 | 2.48 |

*Table 38.* Overall difficulty scores for observation-level perturbation on AgentBenchPlus. Higher scores indicate greater difficulty.

| Condition | ALFWorld | DataBase | KnowledgeGraph | Mind2Web | OperatingSystem | WebShop |
|---|---|---|---|---|---|---|
| Origin | 2.80 | 2.76 | 2.20 | 2.03 | 2.52 | 2.35 |
| Observation-perturb | 2.77 | 2.81 | 2.17 | 2.01 | 2.61 | 2.41 |

*Table 39.* Overall difficulty scores for observation-level perturbation on AgentGymPlus. Higher scores indicate greater difficulty.

| Condition | ALFWorld | BabyAI | MAZE | Movie | SciWorld | SQLGym | TextCraft | Weather | WebShop | Wordle |
|---|---|---|---|---|---|---|---|---|---|---|
| Origin | 4.80 | 4.67 | 3.40 | 2.00 | 4.77 | 4.80 | 4.45 | 2.00 | 2.06 | 3.44 |
| Observation-perturb | 4.94 | 4.56 | 3.56 | 2.00 | 4.65 | 4.84 | 4.37 | 2.05 | 2.06 | 3.32 |

*Table 40.* Performance of non-TrajSFT `Qwen3-8B` under observation-level perturbation on AgentGymPlus.

| Condition | BabyAI | MAZE | SciWorld | SQLGym | TextCraft | Weather | WebShop | Wordle |
|---|---|---|---|---|---|---|---|---|
| Origin | 51.2 | 76.0 | 57.1 | 10.5 | 73.0 | 65.0 | 12.9 | 92.0 |
| Observation-perturb | 59.5 | 88.0 | 63.5 | 13.5 | 68.0 | 50.0 | 13.4 | 96.0 |

*Table 41.* Performance of non-TrajSFT `Qwen3-8B` under observation-level perturbation on AgentBenchPlus.

| Condition | ALFWorld | DataBase | KnowledgeGraph | OperatingSystem | WebShop | Mind2Web |
|---|---|---|---|---|---|---|
| Origin | 20.0 | 26.0 | 20.1 | 8.2 | 17.6 | 22.0 |
| Observation-perturb | 25.0 | 21.2 | 18.8 | 12.5 | 18.1 | 20.2 |

*Table 42.* Performance of trajectory-SFT `Qwen3-8B` under observation-level perturbation on AgentGymPlus.

| Condition | BabyAI | MAZE | SciWorld | SQLGym | TextCraft | Weather | WebShop | Wordle |
|---|---|---|---|---|---|---|---|---|
| Origin | 80.9 | 36.0 | 84.5 | 7.5 | 73.0 | 60.0 | 86.1 | 8.0 |
| Observation-perturb | 79.8 | 20.0 | 83.0 | 9.5 | 69.0 | 45.0 | 86.4 | 20.0 |

*Table 43.* Performance of trajectory-SFT `Qwen3-8B` under observation-level perturbation on AgentBenchPlus.

| Condition | ALFWorld | DataBase | KnowledgeGraph | OperatingSystem | WebShop | Mind2Web |
|---|---|---|---|---|---|---|
| Origin | 30.0 | 38.0 | 14.5 | 6.94 | 13.6 | 6.31 |
| Observation-perturb | 25.0 | 36.3 | 10.9 | 7.64 | 18.8 | 7.40 |

*Table 44.* Diversity-augmented trajectory training on AgentBenchPlus. We vary the proportion of diverse-interface trajectories in the training data and report original-interface performance, perturbed-interface performance, and IR.

| Env. | Ratio | Origin | Perturb-1 | Perturb-2 | IR |
|---|---|---|---|---|---|
| DataBase | 25% | 41.3 | 22.0 | 19.7 | 4.18 |
| | 50% | 38.3 | 33.7 | 32.7 | 2.34 |
| | 75% | 37.3 | 36.0 | 34.7 | 1.16 |
| OperatingSystem | 25% | 6.94 | 5.26 | 42.3 | 2.31 |
| | 50% | 9.72 | 4.86 | 6.25 | 1.22 |
| | 75% | 13.8 | 14.6 | 13.2 | 0.93 |

*Table 45.* Diversity-augmented trajectory training on AgentGymPlus. We vary the proportion of diverse-interface trajectories in the training data and report original-interface performance, perturbed-interface performance, and IR.

| Env. | Ratio | Origin | Perturb-1 | Perturb-2 | IR |
|---|---|---|---|---|---|
| ALFWorld | 25% | 31.0 | 30.5 | 27.5 | 1.05 |
| | 50% | 26.0 | 24.0 | 18.5 | 1.11 |
| | 75% | 26.5 | 35.0 | 28.5 | 1.02 |
| SciWorld | 25% | 69.41 | 61.17 | 62.07 | 0.97 |
| | 50% | 74.58 | 59.36 | 64.63 | 0.93 |
| | 75% | 74.34 | 61.37 | 58.45 | 0.94 |

