# OpenReview forum: "What Do Agents Learn from Trajectory-SFT: Semantics or Interfaces?"
_ICML.cc/2026/Conference — ICML 2026 spotlight_

### Official Review · Reviewer_v3rf · 2026-02-20

**Soundness:** 2
**Presentation:** 3
**Significance:** 2
**Originality:** 2
**Overall Recommendation:** 5
**Confidence:** 3

**Summary:**

This paper propose PIPE to diagnosis the effectiveness of SFT where the memorization of interface API shortcut is outperforming the true semantic tool-use capabilities. The proposed PIPE framework applies to 16 environments including AgentBench and AgentGym and demonstrates that the fine-tuned models are amplifying the interface shortcutting phenomenon. The newly introduced interface reliance (IR) metric might serve as a measurement in mimicking expert trajectories and or in RLVR.

**Compliance With Llm Reviewing Policy:**

Affirmed.

**Final Justification:**

See acknowledgement

**Key Questions For Authors:**

- See above.

**Limitations:**

yes

**Strengths And Weaknesses:**

### Summary Of Strengths
- The findings of the paper are reasonable and intuitive.
- The paper is generally well-written and easy to follow.

### Summary Of Weaknesses
- The two kinds of perturbation families (Synonym-based perturbation and Symbol-based perturbation) appear too 'straightforward' to the reviewer because nowadays many fine-tuning methods already adopt such augmentation tricks to tackle against the overfitting problems of LLMs after SFT. Therefore, it might devalue the depth and effectiveness of the proposed method if only the two kinds of perturbation are considered. In addition, it remains unknown if the SFT techniques investigated in the paper already adopt those augmentation tricks.
- The score metric (sec 3.3) defined by the task success rate alone is sort of simple and might not benefit the discussion on the variation of agentic tool use patterns before and after SFT.
- The authors are encouraged to go one step forward by investigating: 1) whether the current RLVR paradigm can alleviate such phenomenon; and 2) whether we could use the proposed interface reliance metric to: 1) filter expert trajectories and fine-tune LLMs for generalizability concerns; 2) serve as a RLVR reward (together with outcome reward) for polishing the LLM agents via RL. Such newly added experiments would greatly broaden the audience scope and deepen the current studies for development of LLM agnets in the future.

---

> ### Author Rebuttal · Authors · 2026-03-31
>
> We thank the reviewer for these constructive suggestions.
>
> ### W1: The perturbation families may seem too simple
>
> We agree that the two perturbation families in our current paper are simple in form. However, this simplicity is deliberate rather than accidental. Our goal is not to propose a new training-time data augmentation method, but to build a **minimal, controlled evaluation-time probe** that introduces as few confounding factors as possible. For this reason, we change only the surface realization of the interface while keeping the task goal, backend execution, reward, and observations unchanged.
>
> This design choice is important for interpretability: if an agent degrades noticeably even under such minimal rewrites, then the original benchmark gain likely contains a non-trivial dependence on interface surface form, which deserves to be measured explicitly. In other words, the simplicity of the perturbation is part of what makes the diagnostic meaningful.
>
> Regarding whether some of the trajectory-SFT pipelines studied in the paper already use augmentation tricks of this kind: to the best of our knowledge, the specific task-specific post-training settings we evaluate do not explicitly optimize for robustness to alternative interface realizations in the controlled way we test here. We will make this point clearer in the revision and discuss it more explicitly as a limitation and scope condition.
>
> ### W2: Success rate alone is too simple to characterize tool-use patterns
>
> We completely agree, and this is precisely one of the central motivations of our work. Our proposal is not to replace success rate with a single new scalar score. Instead, `PIPE` augments the original benchmark protocol by evaluating under `Origin`, `Perturb-1`, and `Perturb-2`, and then examining the transfer gap across these conditions. In addition, we introduce `IR` to quantify preference for training-time aliases.
>
> Thus, the real object of study in the paper is not "success rate alone," but the combination of:
>
> - original score,
> - perturbed scores,
> - the performance gap across interface conditions, and
> - `IR`.
>
> Together, these signals reveal changes in tool-use behavior that standard evaluation cannot capture. The partial-transfer pattern, repeated invocation of legacy actions in failure cases, and elevated `IR` in some environments are exactly the kinds of behaviors that would be invisible if one only looked at the original benchmark score.
>
> ### W3 / Questions on RLVR and training use of IR
>
> We appreciate these forward-looking suggestions very much. We agree that the following are important next-step questions:
>
> 1. whether RL-based post-training could alleviate interface shortcutting;
> 2. whether `IR` could be used to filter trajectories for generalization;
> 3. whether `IR`-inspired signals could be incorporated into RL rewards.
>
> At the same time, we would like to clarify the scope of the present paper. Our main contribution is to **identify and diagnose** interface reliance under trajectory-SFT, rather than to propose a new training framework that removes it. We therefore see these directions as natural extensions of our findings rather than claims already established by the current paper.
>
> On the RL question, we believe the answer could plausibly be yes, but it requires dedicated validation. In many practical LLM-agent pipelines, RL does not start from scratch but rather builds on an SFT-initialized policy. It is therefore a natural and important next question to ask whether a later RL stage can reduce the interface reliance introduced during the SFT stage. Intuitively, if RL encourages stable behavior across multiple equivalent interface realizations, it may reduce dependence on a single training-time interface. But this cannot be concluded from our current experiments alone, and we will present it explicitly as future work.
>
> Regarding `IR`, we currently view it primarily as a **post-training diagnostic metric**, not as a quantity that can be directly inserted into the training objective without modification. `IR` is defined through a counterbalanced dual-alias evaluation protocol and is therefore an environment-level statistic, rather than a per-step reward that can be optimized directly. That said, its underlying idea can still inform training design. For example, if a model shows high `IR`, this suggests the need to increase interface diversity, strengthen per-tool semantic supervision, or expose the model to multiple equivalent interface realizations during training. Similarly, one could imagine RL rollouts where the same function is available under multiple aliases, together with regularization or reward terms that discourage excessive preference for the training-time alias. We believe these are promising future directions and will discuss them more clearly in the final version.

---

> > ### Author Rebuttal · Reviewer_v3rf · 2026-04-03
> >
> > The authors resolved my concerns and given the research scope, I would increase the score to weak accept.
> > If further experiments on follow-up studies are provided, I would recommend accept.

---

> > > ### Author Response · Authors · 2026-04-06
> > >
> > > Thank you again for your encouraging follow-up and for pointing us to the RL/RLVR direction.
> > >
> > > Following your suggestion, we conducted an additional targeted experiment to directly test whether a post-SFT RL stage can mitigate the interface reliance diagnosed by PIPE. Due to the rebuttal time constraint, we focused on two AgentGym environments that are both supported by the AgentGym-RL framework and show clear post-SFT sensitivity under PIPE in our main paper: SciWorld and BabyAI. Since the current AgentGym-RL framework does not yet support Qwen3, we use Qwen2.5 as the base model in this follow-up study.
> > >
> > > We first train the agent with trajectory-SFT, and then continue training from the SFT-initialized policy with RL. We evaluate the resulting models under the same PIPE protocol, reporting performance on the original interface, the two perturbed interfaces, and IR:
> > >
> > > | Sciworld | Origin-env | Perturb1 | Perturb2 | IR |
> > > |---|---:|---:|---:|---:|
> > > | Origin-model | 43.890 | 44.785 | 45.830 | 0.689 |
> > > | SFT | 79.093 | 55.795 | 55.070 | 9.110 |
> > > | SFT+RL | 89.737 | 62.703 | 57.487 | 7.557 |
> > >
> > > | BabyAI | Origin-env | Perturb1 | Perturb2 | IR |
> > > |---|---:|---:|---:|---:|
> > > | Origin-model | 62.7 | 64.9 | 69.3 | 0.853 |
> > > | SFT | 74.4 | 77.1 | 75.8 | 1.543 |
> > > | SFT+RL | 89.1 | 86.9 | 66.7 | 1.366 |
> > >
> > > Several observations are notable.
> > >
> > > First, before trajectory-SFT, the base model shows little evidence of interface reliance: performance is broadly stable across original and perturbed interfaces, and IR is below 1 in both environments.
> > >
> > > Second, after trajectory-SFT, IR increases substantially in both environments (e.g., from 0.689 to 9.110 in SciWorld), consistent with our main paper’s conclusion that trajectory-SFT can amplify dependence on training-seen interface realizations. In SciWorld, this also coincides with a clear drop from the original interface to the perturbed interfaces.
> > >
> > > Third, after adding RL on top of the SFT policy, we observe a partial mitigation pattern. In both environments, IR decreases relative to SFT alone (SciWorld: 9.110 → 7.557; BabyAI: 1.543 → 1.366). In SciWorld, performance on both perturbed interfaces also improves in absolute terms (55.795/55.070 → 62.703/57.487), suggesting that RL can improve robustness to interface perturbations. In BabyAI, the effect is more mixed: performance improves on Perturb1 but drops on Perturb2, while IR still decreases.
> > >
> > > Therefore, these preliminary results provide initial evidence that RL can partially alleviate certain forms of interface reliance introduced by trajectory-SFT, although the effect is not yet uniform across environments and metrics. We thus view this as supportive—but still limited—evidence for the reviewer’s proposed follow-up direction.
> > >
> > > More broadly, this result suggests that PIPE/IR may be useful not only as post-training diagnostics, but also as tools for analyzing whether later optimization stages such as RL reduce shortcutting. This is well aligned with the reviewer’s suggestion that interface-reliance-aware signals could inform future post-SFT training strategies.
> > >
> > > Taken together, these results provide preliminary support for the reviewer’s suggested RL direction. At the same time, we would like to emphasize that this follow-up experiment does not change the scope of the present paper. Our main contribution remains diagnostic: identifying and measuring interface reliance induced by trajectory-SFT through PIPE and IR, rather than proposing or validating a new training recipe to remove it. We therefore view this RL experiment as a targeted, reviewer-motivated extension that offers only initial evidence that the phenomenon identified by PIPE may be partially mitigated by later optimization stages. Its main value, in our view, is to motivate future work on whether and how PIPE/IR can be used to guide post-SFT optimization, which would require much broader and more systematic study beyond the scope of the current submission.

---

### Official Review · Reviewer_d5MA · 2026-03-11

**Soundness:** 3
**Presentation:** 3
**Significance:** 2
**Originality:** 2
**Overall Recommendation:** 4
**Confidence:** 4

**Summary:**

This paper investigates whether trajectory-based supervised fine-tuning (trajectory-SFT) for LLM agents improves genuine semantic tool understanding or merely induces memorization of interface-specific interaction patterns. The authors propose PIPE, a protocol-level evaluation framework that applies minimal, semantics-preserving perturbations to environment interfaces, along with Interface Reliance (IR), a metric quantifying agents' preference for training-time interfaces. Experiments reveal that trajectory-SFT can substantially amplify interface shortcutting, with trained agents degrading sharply under minimal interface rewrites while non-trajectory-trained models remain relatively stable.

**Compliance With Llm Reviewing Policy:**

Affirmed.

**Final Justification:**

The authors' response addressed my concerns; I have adjusted my score to 4.

**Key Questions For Authors:**

- For untrained models, adding Perturb-2 sometimes improves performance, yet intuitively, removing semantic cues (as in Perturb-2) should degrade tool understanding compared to vanilla or Perturb-1 settings. Why does removing semantic information occasionally lead to better performance than preserving it?
- The categorization of models as "Untrained Agents" versus "Trained Agents" may be ambiguous, as all models have undergone extensive instruction tuning. Does "Trained Agents" here refer specifically to specialized post-training on task-specific trajectories? Clarifying this terminology would help readers better interpret the experimental comparisons.
- To what extent can increasing the diversity of SFT data mitigate the interface reliance issue identified in this work? Could the authors provide empirical evidence or theoretical analysis on how data diversity interacts with interface robustness?

**Limitations:**

Please add sections or paragraphs discussing limitations.

**Strengths And Weaknesses:**

**Strengths:**
- The paper is well-organized with clear motivation, methodology, and experimental design that effectively communicates the core research question to readers.
- The proposed PIPE framework and Interface Reliance metric provide valuable diagnostic tools for evaluating agent robustness beyond standard task success rates.

**Weaknesses:**
- The claim that "a substantial portion of trajectory-SFT's benchmark gains stem from interface shortcuts rather than genuine semantic tool understanding" may be overstated. It is well-established that SFT data lacking diversity can lead to overfitting to specific formats; the examples showing trained agents repeatedly attempting to invoke deprecated tool names reflect overfitting to particular tool schemas—a known phenomenon—rather than a novel "interface shortcut" effect. A more precise framing might be: "Trajectory-SFT without diversity augmentation is prone to interface overfitting," rather than attributing the issue to trajectory-SFT as a paradigm.
- The paper assumes that "semantic understanding" is inherently preferable to "interface memorization," but does not adequately discuss the alignment between training objectives and evaluation protocols. From a practical training perspective, if the goal is to improve performance on a specific downstream task with a fixed interface, strong generalization to perturbed interfaces may not be necessary, and diverse tool schemas in SFT data may not be required. Conversely, if training a generalist agent, one would naturally incorporate diverse trajectories covering different tasks and interfaces; thus, the evaluation under interface perturbations has limited applicability depending on the intended use case.

---

> ### Author Rebuttal · Authors · 2026-03-31
>
> Thank for such valuable review.
> # W1:
>
> We agree. It is more accurate to say that, in common task-specific trajectory-SFT settings without explicit diversity augmentation, the model can improve semantic tool-use while also becoming more dependent on training-time interface surface forms. We will therefore revise the wording accordingly and avoid attributing the issue to trajectory-SFT as a paradigm in an unconditional way.
>
> Our goal is not to criticize trajectory-SFT itself, but to provide a more fine-grained diagnostic framework for interpreting benchmark gains. Due to space constraints, please refer to reviewer zVWP's W2
>
> # W2:
>
> We appreciate this point and agree with the reviewer. Our claim is therefore not that semantic understanding is always the only desirable behavior under every deployment assumption.
>
> Rather, our focus is on the *interpretability of benchmark gains*. As illustrated in Figure 1 of the paper, two agents may both produce the same correct action on the original interface, while relying on very different internal mechanisms: one may choose the action based on understanding the task goal and the tool semantics, whereas the other may simply reproduce a high-frequency interface pattern seen during training. Standard benchmark success rates conflate these cases, so identical scores do not imply identical capabilities.
>
> This distinction can also matter in practice when the task changes even if the environment remains similar. By contrast, an agent with stronger semantic tool understanding is more likely to select the action that is appropriate for the new goal. We will clarify in the revision that `PIPE` is intended as a complementary evaluation signal for benchmark interpretation and training-time diagnosis, rather than as a universal requirement for every deployment scenario.
>
> # Q1:
>
> We also noticed this phenomenon. At this stage, we prefer to offer a plausible explanation consistent with the evidence rather than over-interpret it. One possibility is that, for non-task-specifically trained models, the original action names or their synonyms sometimes trigger lexical priors from pretraining that are not always helpful. By removing these lexical cues, `Perturb-2` may force the model to rely more on the functional descriptions instead of superficial word associations, which can occasionally yield slightly better behavior in some environments.
>
> From our perspective, this observation also supports the core point of the paper: the perturbations are not simply a universal "difficulty increase." Instead, they act as controlled probes that reveal different types of interface dependence. The fact that untrained or non-trajectory-trained models often show mixed or near-zero `\Delta` is itself evidence that the perturbation is not equivalent to uniformly making all tasks harder.
>
> # Q2:
>
> We agree and thank the reviewer for pointing this out and will fix it.
>
> # Q3:
>
> To probe this issue, we added a diversity-augmentation experiment. In addition to the original trajectory data, we construct extra trajectories with alternative interface aliases, and vary their ratio in training data as `25%`, `50%`, and `75%`. To avoid directly exposing the same aliases used by `PIPE` at evaluation time, the diverse trajectories use a different naming scheme, namely `{environment_name}_{function_index}`, rather than the evaluation aliases.
>
> We run this experiment on environments where the trajectory-SFT model showed clear performance drops under `PIPE`. The  results on qwen3-8b are:
>
> | Ratio | AgentBench-env | DBBench | OS |
> |---|---|---:|---:|
> | 25% | Origin | 41.3 | 6.94 |
> | | Perturb-1 | 22.0 | 5.26 |
> | | Perturb-2 | 19.7 | 42.3 |
> | | IR | 4.18 | 2.31 |
> | 50% | Origin | 38.3 | 9.72 |
> | | Perturb-1 | 33.7 | 4.86 |
> | | Perturb-2 | 32.7 | 6.25 |
> | | IR | 2.335 | 1.22 |
> | 75% | Origin | 37.3 | 13.8 |
> | | Perturb-1 | 36.0 | 14.6 |
> | | Perturb-2 | 34.7 | 13.2 |
> | | IR | 1.16 | 0.933 |
>
> | Ratio | Setting | ALFWorld | SciWorld |
> |---|---|---:|---:|
> | 25% | Origin | 31.0 | 69.412 |
> | | Perturb-1 | 30.5 | 61.165 |
> | | Perturb-2 | 27.5 | 62.068 |
> | | IR | 1.047 | 0.967 |
> | 50% | Origin | 26.0 | 74.580 |
> | | Perturb-1 | 24.0 | 59.362 |
> | | Perturb-2 | 18.5 | 64.625 |
> | | IR | 1.106 | 0.927 |
> | 75% | Origin | 26.5 | 74.338 |
> | | Perturb-1 | 35.0 | 61.373 |
> | | Perturb-2 | 28.5 | 58.448 |
> | | IR | 1.022 | 0.939 |
>
> As the proportion of diverse trajectories increases, performance on the original interface remains broadly comparable, perturbed-interface performance improves, and `IR` decreases. This supports a more precise conclusion: benchmark gains after trajectory-SFT may contain a non-trivial interface-reliance component, and increasing interface diversity in the training data appears to be a promising mitigation strategy.
>
> We are extending this experiment to additional environments and will include a more complete set of results in the final version.

---

> > ### Author Rebuttal · Reviewer_d5MA · 2026-04-04
> >
> > Thanks for the answer. The author's response is detailed and valuable.

---

### Official Review · Reviewer_pc7i · 2026-03-11

**Soundness:** 3
**Presentation:** 4
**Significance:** 3
**Originality:** 3
**Overall Recommendation:** 5
**Confidence:** 2

**Summary:**

This paper shows an important but usually ignored problem, that is whether the improved benchmark scores achieved through trajectory supervision fine-tuning. Whether it is due to the model truly understanding the semantics of the tools or merely memorizing superficial shortcuts of the interface. They propose PIPE, a protocol-level evaluation augmentation
for diagnosing interface reliance. Also, they introduce Interface Reliance to measure the model's preference for the interface during training.

**Compliance With Llm Reviewing Policy:**

Affirmed.

**Final Justification:**

The authors have provided experiments in the rebuttal that can fully address my concern. Thus, I chose to turn the score to 5.

**Key Questions For Authors:**

I have no further questions for the authors.

**Limitations:**

yes

**Strengths And Weaknesses:**

Strengths:
The paper is technically sound, and the claims are well supported. The idea is very interesting, and the proposed PIPE is very easy to implement. The experiments are very thorough, which consist of many well known dataset and also many popular LLMs. The training details are well listed in the paper, which makes the paper easy to reproduce.

Weaknesses:
The perturbation has some limitations. Right now, the PIPE may focus on the Action names change. In real agent interactions, changes in the interface may also include alterations in the Observation format, changes in the order of parameter passing, etc. The coverage of these dimensions in the paper is slightly insufficient.

---

> ### Author Rebuttal · Authors · 2026-03-31
>
> We sincerely thank the reviewer for this important suggestion.
>
> ### W1: PIPE currently focuses mainly on action-name perturbations
>
> We agree. The current version of `PIPE` focuses on minimal rewrites of `action names`, mainly because this dimension is the easiest to perturb in a controlled way while preserving backend execution, task goals, and overall intrinsic difficulty. That said, we fully agree that real agent interfaces vary along more than just action names. Observation formatting, parameter ordering, and more general invocation schemas are also meaningful interface dimensions.
>
> Motivated by this suggestion, we designed an additional `observation`-level perturbation. Specifically, before each environment feedback we prepend the natural-language prefix:
>
> `Here is the output of the environment:`
>
> We then examine whether such observation-level perturbations produce phenomena similar to action-level interface shortcutting.
>
> To validate that this observation perturbation does not materially change task difficulty, we apply the same two difficulty-control procedures described in our response to Reviewer zVWP: pairwise relative comparison and absolute difficulty scoring. In the pairwise setting, all environments are judged as `Same` in `100%` of the cases, so we omit the full per-environment table for brevity.
>
> The absolute scoring results are:
>
> | AgentBench-env | alfworld | dbbench | knowledgegraph | mind2web | os | webshop |
> |---|---:|---:|---:|---:|---:|---:|
> | Origin | 2.80 | 2.76 | 2.20 | 2.03 | 2.52 | 2.35 |
> | Observation-perturb | 2.77 | 2.81 | 2.17 | 2.01 | 2.61 | 2.41 |
>
> | AgentGym-env | ALFWorld | BabyAI | MAZE | Movie | SciWorld | SQLGym | TextCraft | Weather | WebShop | Wordle |
> |---|---:|---:|---:|---:|---:|---:|---:|---:|---:|---:|
> | Origin | 4.80 | 4.67 | 3.40 | 2.00 | 4.77 | 4.80 | 4.45 | 2.00 | 2.06 | 3.44 |
> | Observation-perturb | 4.94 | 4.56 | 3.56 | 2.00 | 4.65 | 4.84 | 4.37 | 2.05 | 2.06 | 3.32 |
>
> These results again suggest that the perturbation does not systematically increase difficulty.
>
> We then evaluate `Qwen3-8B` before and after task-specific trajectory-SFT under this observation perturbation.
>
> For the untrained `Qwen3-8B`:
>
> | AgentGym-env | BabyAI | Maze | SciWorld | SQLGym | TextCraft | Weather | WebShop | Wordle |
> |---|---:|---:|---:|---:|---:|---:|---:|---:|
> | Origin | 51.2 | 76.0 | 57.1 | 10.5 | 73.0 | 65.0 | 12.9 | 92.0 |
> | Observation-perturb | 59.5 | 88.0 | 63.5 | 13.5 | 68.0 | 50.0 | 13.4 | 96.0 |
>
> | AgentBench-env | ALFWorld | DBBench | KnowledgeGraph | OS | WebShop | Mind2Web |
> |---|---:|---:|---:|---:|---:|---:|
> | Origin | 20.0 | 26.0 | 20.1 | 8.2 | 17.6 | 22.0 |
> | Observation-perturb | 25.0 | 21.2 | 18.8 | 12.5 | 18.1 | 20.2 |
>
> For the trajectory-trained `Qwen3-8B`:
>
> | AgentGym-env | BabyAI | Maze | SciWorld | SQLGym | TextCraft | Weather | WebShop | Wordle |
> |---|---:|---:|---:|---:|---:|---:|---:|---:|
> | Origin | 80.9 | 36.0 | 84.5 | 7.5 | 73.0 | 60.0 | 86.1 | 8.0 |
> | Observation-perturb | 79.8 | 20.0 | 83.0 | 9.5 | 69.0 | 45.0 | 86.4 | 20.0 |
>
> | AgentBench-env | ALFWorld | DBBench | KnowledgeGraph | OS | WebShop | Mind2Web |
> |---|---:|---:|---:|---:|---:|---:|
> | Origin | 30.0 | 38.0 | 14.5 | 6.94 | 13.6 | 6.31 |
> | Observation-perturb | 25.0 | 36.3 | 10.9 | 7.64 | 18.8 | 7.4 |
>
> These results suggest that, in most environments, observation perturbation does not cause the same pronounced degradation as action-name perturbation. In other words, under the simple observation rewrite we tested, the model does not exhibit a comparably strong form of brittleness.
>
> A plausible explanation is that trajectory-SFT directly supervises action generation, not the exact surface form of observation text, so models may be less dependent on fixed observation formatting than on stable action names. Another possible reason is that environment feedback is naturally more diverse, while action names are typically highly stable across training trajectories and are therefore easier to memorize as surface patterns.
>
> We fully agree that `observation format`, parameter ordering, and more general invocation-schema changes are important extensions. In the revision, we will present the current work more explicitly as a study of **minimal interface perturbation**, and we will broaden the discussion of these additional interface dimensions.

---

> > ### Author Rebuttal · Reviewer_pc7i · 2026-04-02
> >
> > Thanks for your answer; my concern is fully resolved.

---

### Official Review · Reviewer_zVWP · 2026-03-12

**Soundness:** 3
**Presentation:** 3
**Significance:** 3
**Originality:** 3
**Overall Recommendation:** 4
**Confidence:** 4

**Summary:**

This paper argues that standard agent benchmarks confound two distinct sources of success for trajectory-SFT agents: genuine semantic tool understanding and memorization of interface-specific action patterns. To diagnose this, the authors propose PIPE, an evaluation protocol that minimally perturbs action interfaces while preserving task semantics and executable behavior, and Interface Reliance (IR), a metric that quantifies preference for training-time action names. Across 16 environments from AgentBench and AgentGym, the paper reports that trajectory-SFT often yields strong gains on the original interface but substantially worse performance under perturbed interfaces, while non-trajectory-trained models are comparatively stable. The paper also includes failure cases and a training-dynamics case study, suggesting that standard benchmark success alone can mask brittle interface shortcutting.

**Compliance With Llm Reviewing Policy:**

Affirmed.

**Final Justification:**

The clarification and new experiments address my concerns, so I will keep my positive score.

**Key Questions For Authors:**

1. Can the authors provide stronger evidence that PIPE perturbations preserve task difficulty beyond the behavior of a few untrained or API-based models?

**Limitations:**

The writing might need to be more careful about what exactly has been established

**Strengths And Weaknesses:**

Strengths:
1. The paper addresses a timely and important evaluation problem for LLM agents.

2. The proposed intervention is simple and elegant. PIPE changes only action names/interface realizations while preserving descriptions and underlying execution, which makes it a plausible diagnostic for interface dependence rather than a wholesale distribution shift. The distinction between synonym-based and symbol-based perturbations is also sensible.

3. The empirical coverage is fairly broad. The evaluation spans AgentBench and AgentGym, includes open-source trajectory-trained agents, newly fine-tuned models on AgentGym trajectories, and several API-based models. The reported effects are large in some environments, especially WebShop, ALFWorld, SciWorld, and DataBase, which makes the phenomenon hard to dismiss as noise.

Weaknesses:

1. The paper might sometimes overstate what PIPE can prove. A drop under interface perturbation is certainly consistent with interface shortcutting, but it is not a clean identification of “semantic learning” versus “interface memorization.” Renaming actions can still change usability, readability, or prompt salience in subtle ways, even if underlying execution is preserved. The authors argue that perturbations do not uniformly increase intrinsic difficulty because untrained models are often stable, but this is only indirect evidence.

2. Claims like trajectory-SFT “inherently induces” interface shortcuts feel too broad given that the observed effect is highly environment-dependent and some trained agents remain fairly robust. The appendix snippet even argues that shortcutting depends on environment structure, tool diversity, and supervision density, which suggests a more nuanced story than the main claims sometimes present.

3. The perturbation design needs stronger validation. The paper says the aliases were “carefully designed” not to increase task difficulty, but this is a critical assumption and deserves more direct evidence in the main paper, perhaps via human validation or an oracle parser-based control. Right now, too much of this burden is pushed to appendices.

---

> ### Author Rebuttal · Authors · 2026-03-31
>
> We sincerely appreciate the reviewer's thoughtful comments. We agree that some statements in the current version are too strong, and we will revise them carefully.
>
> ### W1: PIPE may overclaim what it identifies
>
> We agree. Our goal is not to claim that `PIPE` can *cleanly* or *fully* separate semantic learning from interface memorization in a strict causal sense. Rather, `PIPE` provides a finer-grained diagnostic signal than standard benchmark evaluation: it tests whether gains obtained on the original interface transfer to minimally rewritten but behavior-preserving interfaces. A drop under perturbation is therefore evidence *consistent with* interface shortcutting, not a perfect proof of its absence or presence.
>
> We will revise the paper to make this framing explicit. In particular, we will avoid wording that suggests strict identification, and instead emphasize that `PIPE` is a simple, plug-and-play evaluation augmentation that helps reveal whether benchmark gains are robust to small interface rewrites.
>
> ### W2: Claims about trajectory-SFT are too broad
>
> We agree with this concern as well. Our intention is not to argue that trajectory-SFT as a paradigm should be rejected. The more precise claim is that, in common task-specific trajectory-SFT settings, training can improve semantic capability *and* simultaneously amplify dependence on training-seen interface realizations, with the extent of this effect being strongly environment-dependent.
>
> We will therefore revise expressions such as "inherently induces" to more careful formulations such as "can induce" or "can amplify." We will also make clearer that interface shortcutting is not an unavoidable property of trajectory-SFT. In fact, our case study on training-time diagnostics already suggests that, in some environments, continued training and better checkpoint selection can reduce `IR` and partially recover perturbed-interface performance.
>
> ### W3 / Q1: Stronger evidence that the perturbations preserve task difficulty
>
> We appreciate this important suggestion, and we have added two new difficulty-control experiments using an LLM-as-judge setup. In both experiments, we concatenate the environment description and task description into a full task specification, and ask the evaluator to judge difficulty under the assumption that the evaluated agent is a competent agent that can read action descriptions. Due to space constraints we do not include the full prompt here, but we will add it in the final version. We use `gpt-5.1` as the evaluator.
>
> #### Experiment 1: Pairwise relative comparison
>
> We compare `Origin vs. Perturb-1` and `Origin vs. Perturb-2` in a paired fashion. The evaluator only judges whether, assuming unchanged backend dynamics, task goals, and action semantics, the interface rewrite makes the task `Easier`, `Same`, or `Harder`.
>
> We run this evaluation on all environments in `AgentBenchPlus` and `AgentGymPlus`. The evaluator returns `Same` for almost all environments; the only small deviation appears in the `Movie` environment of `AgentGymPlus`, where the agreement with `Same` is still `95%`. This suggests that, from the perspective of a strong evaluator, the rewritten interfaces do not materially change task difficulty in almost all cases.
>
> #### Experiment 2: Absolute difficulty scoring
>
> We also score `Origin`, `Perturb-1`, and `Perturb-2` separately rather than comparing them jointly. The evaluator rates each condition on a 1-5 scale along dimensions including clarity of the interface/task description, interaction or reasoning complexity, and overall difficulty, where higher means harder.
>
> Average overall scores are as follows:
>
> | AgentBench-env | alfworld | dbbench | knowledgegraph | mind2web | os | webshop |
> |---|---:|---:|---:|---:|---:|---:|
> | Origin | 2.80 | 2.76 | 2.20 | 2.03 | 2.52 | 2.35 |
> | Perturb-1 | 2.90 | 2.69 | 2.10 | 2.07 | 2.42 | 2.40 |
> | Perturb-2 | 2.95 | 2.74 | 2.25 | 2.06 | 2.74 | 2.48 |
>
> | AgentGym-env | ALFWorld | BabyAI | MAZE | Movie | SciWorld | SQLGym | TextCraft | Weather | WebShop | Wordle |
> |---|---:|---:|---:|---:|---:|---:|---:|---:|---:|---:|
> | Origin | 4.80 | 4.67 | 3.40 | 2.00 | 4.77 | 4.80 | 4.45 | 2.00 | 2.06 | 3.44 |
> | Perturb-1 | 4.87 | 4.58 | 3.60 | 2.00 | 4.69 | 4.76 | 4.38 | 2.15 | 2.18 | 3.36 |
> | Perturb-2 | 4.78 | 4.58 | 3.40 | 2.00 | 4.86 | 4.82 | 4.50 | 2.45 | 2.01 | 3.64 |
>
> These scores do not show a systematic shift toward higher difficulty under perturbation. While we did not have time to conduct human-expert validation before rebuttal, we agree that this would be valuable and will include stronger validation in the final version. We also agree that this material should be moved from the appendix into the main paper.

---

> > ### Author Rebuttal · Reviewer_zVWP · 2026-04-04
> >
> > The clarification and new experiments address my concerns.

---

### Decision · Program_Chairs · 2026-04-30

**Decision:**

Accept (spotlight)

**Comment:**

The paper tackles a critical and timely question in the evaluation of LLM agents: do improvements from trajectory-based supervised fine-tuning (trajectory-SFT) stem from genuine semantic tool understanding or superficial interface memorization? The authors introduce PIPE (Perturb Interface Protocol for Evaluation), a diagnostic framework that applies minimal, behavior-preserving rewrites to environment interfaces (e.g., renaming actions to synonyms or random symbols). They also propose Interface Reliance (IR), a metric to quantify preference for training-time aliases. Experiments across 16 environments from AgentBench and AgentGym reveal that while non-trajectory-trained models are stable under these shifts, trajectory-SFT agents often suffer sharp performance drops, indicating a strong reliance on "interface shortcuts."

Strengths:
- The reviewers found the core research question highly relevant. The proposed PIPE intervention is elegant—changing only surface realizations while keeping task semantics and backend execution constant provides a clean diagnostic tool.
- The authors evaluated a broad range of models (open-source, newly fine-tuned, and API-based) across a diverse set of environments. The large performance drops observed (e.g., in WebShop and SciWorld) make the findings hard to dismiss as noise.
- The case study on training dynamics demonstrates that standard benchmark scores can be non-monotonic or misleading, whereas PIPE and IR provide a more faithful signal for model selection and checkpointing.


The rebuttal was exceptionally productive, with all reviewers moving to an accepting stance.
- In response to concerns about whether perturbations increased intrinsic task difficulty, the authors provided gpt-5.1 judge experiments and absolute difficulty scoring, showing no systematic shift in task complexity.
- The authors clarified that they do not claim trajectory-SFT is "inherently" flawed, but rather that common task-specific SFT can amplify interface reliance. They added a diversity-augmentation experiment during the rebuttal, showing that including alternative aliases during training can successfully mitigate shortcutting.
- Following reviewer suggestions, a targeted experiment showed that Reinforcement Learning (RL) on top of SFT policies can partially reduce IR and improve robustness, opening a promising direction for future work.
- The authors even included a preliminary study on observation-level perturbations (e.g., adding natural language prefixes), finding that models are generally more robust to observation changes than action-name changes.

The paper makes a solid diagnostic contribution that will likely influence how the community evaluates agentic robustness. The rebuttal addressed all technical concerns and added valuable evidence on how to mitigate the identified shortcuts.